# Ca^2+^ Signalling and Hypoxia/Acidic Tumour Microenvironment Interplay in Tumour Progression

**DOI:** 10.3390/ijms23137377

**Published:** 2022-07-02

**Authors:** Madelaine Magalì Audero, Natalia Prevarskaya, Alessandra Fiorio Pla

**Affiliations:** 1U1003—PHYCEL—Laboratoire de Physiologie Cellulaire, Inserm, University of Lille, Villeneuve d’Ascq, 59000 Lille, France; madelaine.audero@univ-lille.fr (M.M.A.); natacha.prevarskaya@univ-lille.fr (N.P.); 2Laboratory of Cellular and Molecular Angiogenesis, Department of Life Sciences and Systems Biology, University of Turin, 10123 Turin, Italy

**Keywords:** Ca^2+^ signalling, TRP channels, SOC channels, PIEZO channels, tumour acidic microenvironment, hypoxia, tumour progression

## Abstract

Solid tumours are characterised by an altered microenvironment (TME) from the physicochemical point of view, displaying a highly hypoxic and acidic interstitial fluid. Hypoxia results from uncontrolled proliferation, aberrant vascularization and altered cancer cell metabolism. Tumour cellular apparatus adapts to hypoxia by altering its metabolism and behaviour, increasing its migratory and metastatic abilities by the acquisition of a mesenchymal phenotype and selection of aggressive tumour cell clones. Extracellular acidosis is considered a cancer hallmark, acting as a driver of cancer aggressiveness by promoting tumour metastasis and chemoresistance via the selection of more aggressive cell phenotypes, although the underlying mechanism is still not clear. In this context, Ca^2+^ channels represent good target candidates due to their ability to integrate signals from the TME. Ca^2+^ channels are pH and hypoxia sensors and alterations in Ca^2+^ homeostasis in cancer progression and vascularization have been extensively reported. In the present review, we present an up-to-date and critical view on Ca^2+^ permeable ion channels, with a major focus on TRPs, SOCs and PIEZO channels, which are modulated by tumour hypoxia and acidosis, as well as the consequent role of the altered Ca^2+^ signals on cancer progression hallmarks. We believe that a deeper comprehension of the Ca^2+^ signalling and acidic pH/hypoxia interplay will break new ground for the discovery of alternative and attractive therapeutic targets.

## 1. Introduction

### 1.1. Cancer Microenvironment: Focus on Tumour Acidic pH_e_ and Hypoxia

Solid tumours are characterised by a dynamic microenvironment constituted by a variety of different non-cellular components, such as the extracellular matrix (ECM) components, circulating free DNA, and cell components, such as aberrant blood vessels, immune cells, tumour-associated fibroblasts (TAFs), endothelial cells, macrophages, pericytes, among others. In turn, the interaction with chemical and physical cues (hypoxia, tumour acidosis, high tumour interstitial stiffness), originates finally a peculiar chemical and physical environment that supports cancer progression [1].

A common feature of almost all advanced solid cancers is the presence of transient or permanent acidic and hypoxic tumour regions, which are a direct outcome of the cancer cells’ metabolic pathways rearrangement, supporting their uncontrolled proliferation. This leads to a significant increase in cancer cells’ anabolic activity and a reduction in the catabolic one, promoting the synthesis of amino acids, nucleotides, and lipids to back their growth. Indeed, according to Otto Warburg’s work [2], cancer cells are characterised by an enhanced glycolytic breakdown of glucose to pyruvate and consequent NADPH and ATP production with respect to healthy cells, even in presence of oxygen and even considering the lower energy yield of pyruvate fermentation compared to oxidative respiration. Nevertheless, it is important to underline that not all cancer cells are characterized by the Warburg effect, but it has been observed that cancer cells may have an opposite phenotype, with an increased mitochondrial oxidative activity [3]. The importance of mitochondrial activity in cancer cells is also explained by the oxidative phosphorylation increase observed when the Warburg effect is inhibited in cancer cells [3,4,5].

The metabolic rearrangement and fermentation of pyruvate resulting from glycolysis lead to high production of lactic acid, acidifying the intracellular environment. Hydrolysis of ATP also determines the release of protons (H^+^) in the intracellular space, contributing to its acidification [6,7,8]. Protonation has a severe negative impact on several enzymes and lipids, leading to a potential risk of impinging several cellular processes, including cell metabolism. It is therefore not surprising that cells use different systems to maintain the intracellular pH (pH_i_) within the physiological range of about 7.2. In cancer cells, a slightly more alkaline intracellular pH (pH 7.4) has been observed, and evidence showed that this slight difference in pH_i_ between cancer cells and healthy cells promotes some of the hallmarks of cancer, such as cell death escape and proliferation [9,10,11,12]. To maintain this pH_i_ value, transformed cells have at their disposal an arsenal of overexpressed transporter proteins and pumps for protons and lactic acid extrusion to the extracellular milieu, resulting in its acidification. Examples of this transport system comprise monocarboxylate transporters (MCTs), major players in the transmembrane lactate trafficking, Na^+^/H^+^ antiporters (NHE), vacuolar H^+^ ATPases, and carbonic anhydrases (CAs), mainly CAIX and CAXII, which role in cancer progression is well documented [13,14,15,16,17]. Acidic interstitial fluids are not only the result of the presence of lactic acid and protons but also of the CO_2_ derived from the cell respiration process in more oxygenated areas. CO_2_ can passively diffuse through the plasma membrane (PM), or it can be reversibly hydrated to HCO_3_^−^ by the transmembrane Carbonic Anhydrase IX (CAIX) exofacial site and released in the tumour microenvironment with protons. Na^+^/HCO_3_^−^ cotransporters (NBCs) in the proximity of CAIX can mediate HCO_3_^−^ influx for sustaining intracellular buffering, titrating cytosolic H^+^ [18,19]. It has to be noticed that the pH_i_ of cancer cells can drop significantly in presence of a strong acidic pH_e_, giving rise to a heterogeneous pH_i_ landscape (due to the TME), where cancer cells resident in acidic regions will present quite a low pH_i_, while cancer cells occupying moderate acidic pH_e_ areas will show a moderate alkaline pH_i_ [8]. It is therefore important to consider the tumour microenvironmental complexity to understand how cancer cells adapt to it, in order to possibly find new therapeutical targets. In addition to metabolic rearrangement, tumour acidosis can be further boosted by tumour-associated hypoxia, which leads to higher glycolytic cell metabolism rates. Hypoxia occurs in the context of tumours vascularized by insufficient vessels and/or vessels characterised by a poor capacity to diffuse oxygen and nutrients and to remove the metabolic waste products due to an altered process of angiogenesis, which leads to the formation of aberrant and dysfunctional vessels. Hypoxia is also the result of an increased oxygen demand from highly proliferating areas of the tumour, leading to intratumour hypoxia heterogeneity, with subregions of the tumour characterised by different oxygen concentrations and consumption. The irregular exposure to oxygen fluctuations is associated with adaptive mechanisms set in motion by cancer cells in order to promote their survival in that hostile environment. Indeed, hypoxia adaptation is linked to increased genomic instability and tumourigenesis [20] and to more aggressive cancer phenotypes in terms of tumour growth, drug and cell death resistance, angiogenesis and enhanced metastasis [21].

Hypoxia adaptation processes are initiated by a series of transcription factors belonging to the hypoxia-inducible factor family, in particular hypoxia-inducible factor 1 (HIF-1), which determines a gene expression reprogramming that affects cancer cell metabolism and processes which sustain its progression. HIF-1 is a heterodimer protein constituted by HIF-1α and HIF-1β and this complex is not present in normoxic conditions. Although the β subunit is constitutively expressed in all cells, the HIF-1α subunit is present only at low levels in all cells’ cytoplasm due to the presence of two specific proline residues at positions 402 and 564 in the oxygen-dependent degradation (ODD) domain in the α subunit. These residues are hydroxylated by prolyl hydroxylase protein (PHD) in presence of physiological oxygen levels, and this modification targets the subunit to degradation via the ubiquitin-proteosome pathway [22]. However, in hypoxic conditions, PHD is inhibited and therefore Pro402 and Pro564 are not hydroxylated, allowing HIF-1α and HIF-1β to dimerize and translocate to the cell nucleus and the consequent activation of target genes transcription [22]. In addition to low O_2_ concentrations, high intracellular lactate levels and specific growth factors or oncogenes can stabilise HIF-1α as well, leading to the activation of HIF-1α-target genes [23,24]. “Hypoxia-adaptive” responsive genes include glucose transporters, such as GLUT1/3, enzymes involved in anaerobic glycolysis, such as lactate dehydrogenase-A (LDHA), aldolase (ALDA), phosphoglycerate kinase-1 (PGK1), enolase (ENOL) and phosphofructokinase-1 (PFK-1), pyruvate dehydrogenase kinase 1 (PDK1), with consequent suppression of mitochondrial oxidative phosphorylation system (OXPHOS).

The hypoxia response is not limited to glycolytic flux, as it also enhances the expression of VEGF and other pro-angiogenic factors and promotes tumour progression by inducing epithelial–mesenchymal transition (EMT) [25], cell survival in moderate hypoxic conditions via autophagy, cell death via the same mechanism but in presence of severe hypoxic conditions [26], and by promoting cancer cells’ invasion and metastasis in acute hypoxic conditions (from minutes up to 72 h exposure in vitro). In particular hypoxia-mediated cell invasion is achieved by sustaining the mesenchymal phenotype, the expression of different metalloproteases (MMPs), lysyl oxidase (LOX), connective tissue growth factor (CTGF) and CAIX, NHE1 and MCTs [21], which contribute to pH regulation and enhance the acidification of the tumour microenvironment.

Hypoxia also plays a key role in chemoresistance, as reduced oxygen availability can affect not only drug delivery but also chemotherapeutics activity [27]. Moreover, the tumour hypoxic core is occupied by cancer cells with a hypoxia-induced stem cell-like phenotype, characterized by cell cycle arrest in the G1 phase and a quiescent state, representing a major problem for those chemotherapy agents which target rapidly proliferating cells [27,28,29]. Hypoxia also upregulates multidrug resistance genes [30,31,32].

Hypoxia sustains all these effects by activating different signalling pathways via HIF-1α, such as the Wnt/β-catenin and TGF-β/SMAD pathways, or in a HIF-1α-independent manner, switching on MAPKs, AMPKs, PI3K/Akt/mTOR, NF-κB and Notch signalling transduction pathways [21,25], which are dependent also on intracellular Ca^2+^ ions. All these mechanisms of cell adaptation to hypoxia determine the selection of highly aggressive clones, which pave the way for tumour expansion.

In addition to being affected by the hypoxic tumour microenvironment, acidic pH_e_ has been observed to regulate HIF1α and HIF2α levels under normoxic conditions in glioma cells, promoting cancer stem cell maintenance [33], highlighting the feedback regulation and crosstalk between hypoxia and low pH_e_. Similarly to hypoxia, the acidic tumour microenvironment supports different hallmarks of cancer, such as drug resistance as previously described [34]. In addition, acidic TME plays an important role in immunoreactive processes and inflammation, by promoting the viability and fitness of pro-tumour M2 macrophages with respect to anti-tumour M1 macrophages [35], by inhibiting T and NK cells activation and inducing immune escape [36,37], by inducing a phenotypic shift in macrophages towards a tumour-promoting phenotype [38] and by increasing the tumour-promoting functions of tumour-associated neutrophils [39]. Moreover, acidic pH_e_ fulfils its pro-tumour function through the enhancement of two other important hallmarks: cancer cell invasion and the ability to metastasise [40]. Studies in breast cancer and colon cancer have indeed demonstrated that invasive cell areas co-localise with acidic pH_e_ regions [41], while studies in melanoma cells have shown that acidic pH_e_ exposure increases their invasive abilities in vitro and the formation of pulmonary metastasis in vivo via a low pH_e_-promoted secretion of proteolytic enzymes and pro-angiogenic factors [42].

An explanatory example of the major role of acidosis in cancer progression is given by the unique pancreatic ductal adenocarcinoma (PDAC) microenvironment [43]. The pancreatic duct is a net acid-base transporting epithelium, in which ductal cells secrete bicarbonate into the ductal lumen across the apical membrane. This transport is coupled to the extrusion of an equal amount of acid across the basolateral membrane, thereby physiological pancreatic interstitium is substantially acidic and epithelial cells are exposed to different extracellular pH (pH_e_) values. This process is intermittent in the healthy pancreas and associated with food intake. On the other hand, PDAC has been clearly associated with a hypoxic and acidic microenvironment with a dense desmoplastic stroma [43]. A challenging hypothesis is that, in combination with driver mutations, the alternating, but physiological, pH_e_ landscape in the pancreas and the intrinsic ability of pancreatic epithelial cells to adapt to different pH conditions, may act as a “preconditioning phenomenon” favouring the selection of specific cancer aggressive phenotypes which might promote PDAC arising and/or progression. In other words, once the specific mutations drive the ductal pancreatic cells’ transformation, cells would be already adapted and could even benefit from the adverse pH conditions and the combination of these factors may increase cell fitness to survive and become strongly aggressive in the hostile microenvironment [43].

In this context, it would be important to study the “transportome” alterations that are linked to hypoxia and to pH_e_ or pH_i_ alterations as possible targets for therapies. Indeed, many cancer hallmarks, such as cell proliferation, cell migration, invasion, and apoptosis resistance are driven by altered expression/regulation of ion transport proteins or ion channels, including acid–base transporters and O_2_- and pH-sensitive channels, in particular, Ca^2+^- and hypoxia- and pH-sensitive ion channels [43,44].

### 1.2. Calcium Signalling

Among different ions present in the intra- and extracellular environments, Ca^2+^ ions stand out for their functional importance as second messengers. Ca^2+^ ions have been observed to crosstalk with several cell signalling pathways by promoting different spatio-temporal Ca^2+^ patterns to selectively regulate innumerable physiological cell processes, ranging from cell differentiation, proliferation, migration, and programmed cell death to gene transcription, among others [45,46,47]. Its key role in signal transduction translates into the necessity of tight regulation of intracellular Ca^2+^ homeostasis, maintaining a low cytosolic free Ca^2+^ concentration (100 nM) with respect to the extracellular milieu (>1 mM) through the orchestrated work of several proteins that constitute the so-called Ca^2+^ signalling toolkit, including pumps (Ca^2+^ ATPases PMCA, SERCA) exchangers (Na^+^/Ca^2+^ exchanger NCLX in mitochondria and NCX at the plasma membrane) or uniporters (MCU in mitochondria) and PM and ER Ca^2+^-permeable channels. Calcium signals are modulated in time and space and differences in amplitude, frequency, duration, and location are transduced by cells to activate a specific response.

Thus, considering this multifaced role of Ca^2+^ ions, it is no surprise that alterations in Ca^2+^ homeostasis and Ca^2+^ channels expression and/or activity in cancer progression and vascularisation have been extensively reported by several works [44,48,49,50,51,52,53,54,55,56,57], making ion channels major players in cancer development (“Oncochannelopathies”) [58].

Ca^2+^-permeable channels represent pivotal molecular devices acting as microenvironmental sensors in the context of tumourigenesis and other diseases, being modulated by microenvironmental physicochemical cues, such as hypoxia and acidic pH_e_. Thus, it is possible to speculate that acidic and hypoxic TME and Ca^2+^ signalling may work in synergy for the acquisition of aggressive cancer cell phenotypes. For this reason, a better understanding of the interplay between these players and the remodelling of Ca^2+^ signals induced by tumour acidic pH_e_ and hypoxia and translated to the cancer cells through the activity of Ca^2+^-permeable ion channels and pumps may help to provide further comprehension of the mechanisms of cancer progression and novel putative therapeutic approaches. Indeed, hypoxia is often associated with a rise in intracellular Ca^2+^ levels in several types of cancer, via the upregulation of different Ca^2+^-permeable channels and eventually potentiating cancer hallmarks [59,60] (see Section 2). In addition to hypoxia, acidic pH has been reported to regulate Ca^2+^-permeable ion channels in a direct and indirect way, via specific H^+^ binding sites or via competition with Ca^2+^ ions for binding sites as discussed below in Section 2. Data regarding the effects of pH_i_ on these channels is limited in the literature and refers principally to normal cells, whereas more data are available on the regulation of acidic pH_e_, although different publications demonstrated the high variability of the effects of this microenvironmental parameter among different Ca^2+^ channels [44,57,61] (see also detailed discussion in Section 2).

In the next section, we will present an updated view of the recent literature on the role of Piezo channels, transient receptor potential Ca^2+^-permeable ions channels (TRPs) and the so-called store-operated Ca^2+^ channels (SOCs) on normal and cancer cells, which activity and signalling transduction are directly affected by two features of the tumour microenvironment: hypoxia and acidosis. We will moreover illustrate the Ca^2+^ signalling pathways that may represent potential targets for cancer therapy.

## 2. Hypoxia and Acidic pH_e_-Dependent Regulation of Ca^2+^-Permeable Ion Channels in Normal and Cancer Cells

Hypoxia and acidic pH_e_ regulate the expression and/or activity of several Ca^2+^-permeable channels, which are linked to tumour aggressiveness. Throughout the years, several publications have revealed the role of these two major players in the tumour microenvironment, proving a marked sensitivity to oxygen and pH_e_ of most TRPs, SOCs and Piezo channels, which affects their functionality in different tissues. A detailed description of the updated literature on both normal and cancer cells will be presented in this section and Table 1 and Table 2.

As summarized in Table 1 and Table 2, the effects of tumour acidosis and tumour hypoxia vary significantly between the different calcium-permeable channels, and the information in the literature regarding certain channels is sometimes contradictory or limited to normal cells.

**Table 1 ijms-23-07377-t001:** Ca^2+^-permeable ion channels regulation by acidic pH_e_.

Ion Channel	Cell Type	Methodology	Acidic pH Value and Treatment Time	Effect of Low pH on Channel’s Activity/Expression	Effect of Low pH on Ca^2+^ Signals	Cellular Function	Ref.
**Piezo1**	Piezo1-transiently transfected HEK293 cells	Patch clamp Mn^2+^ quenching assay	pH_e_ 6.3–6.7, acute treatment	Stabilization of inactivated state, both acidic pH_i_ and pH_e_ inhibit channel’s activity	Decreased Ca^2+^ influx	Not assessed	[62]
	Murine pancreatic stellate cells (mPSCs)	Mn^2+^ quenching assay mPSCs spheroids viability and apoptosis assay	pH_e_ 6.6 and pH_i_ 6.77 (obtained by 30 mM propionate) in acute treatment for Mn^2+^ quenching assay, while 24 h long treatment for spheroid histology	Acidic pH_e_ do not modify Piezo1 activity, while intracellular acidification inhibits channel’s activity	Acidic pH_e_ do not modify Ca^2+^ influx, while intracellular acidification decreases Ca^2+^ influx	Acidic pH_e_ (6.6) impairs PSCs spheroid’s integrity and viability, inducing cell apoptosis	[63]
**TRPM2**	Inducible TRPM2-overexpressing HEK293	Patch clamp	External solution with pH 5–8 superfused for 200 s. Internal solution with pH 6 superfused for 100 s; External solution with pH 3.5–6.5 in acute treatment or more prolonged periods (≥2 min)	Extracellular acidification inactivates the channel in a voltage-dependent manner and [H^+^]-dependent manner. Intracellular acidification induces channel closure	Not assessed, but recovery from acidic pH-induced inactivation requires external Ca^2+^ ions	Not assessed	[64]
	Human neutrophils	Patch clamp	External solution with pH 5 in acute treatment	External acidification negatively affects open probability and single-channel conductance, inducing channel closure	Not assessed	Not assessed	[64]
	TRPM2-overexpressing HEK293	Patch clamp	External solution with pH 3.5–6 in acute treatment	External acidification (up to pH 4.5) reversely decreases mean current amplitude in a [H^+^]-dependent manner, decreasing single-channel conductance	Not assessed	Not assessed	[65]
	TRPM2-overexpressing HEK293	Patch clamp	External solution with pH 4.0–6.5. Different time exposition based on protocol (from <10 s to ≥2 min)	Acidic pH_e_ inactivates open channels in an irreversible manner. Exposition to pH_e_ 4–5 negatively affects channel activation.	Not assessed	Not assessed	[66]
	TRPM2-overexpressing HEK293	Patch clamp	External solution with pH 5.5, different exposition times (0, 30, 60, 90, and 120 s)	Irreversible inhibition after ≤60 s exposure	Not assessed	Not assessed	[67]
**TRPM6**	Pig isolated ventricular myocytes	Patch clamp	External solution with pH 5.5 and pH 6.5, ~5–10 min exposition	External acidification decreases channel’s current amplitude in a pH_e_-dependent and voltage-independent manner. The inhibitory effect of acidic pH_e_ is prevented by increasing intracellular pH buffering capacity	Not assessed	Not assessed	[68]
	TRPM6-overexpressing HEK293 cells	Patch clamp	External solution with pH 3–6, ~10 s-long exposition	External acidification increases channel’s current amplitude in a pH_e_-dependent manner	Not assessed	Not assessed	[69]
**TRPM7**	RBL-2H3 cells	Patch clamp	Acidification of intracellular side of membrane with ~200 s long 4–40 mM acetate treatment	Pre-incubation in 40 mM acetate solution inhibits TRPM7 current in a reversible manner	Not assessed	Not assessed	[70]
	TRPM7-overexpressing Chinese Hamster Ovary (CHO-K1) cells	Patch clamp	Internal and external solution with pH 5.6 and variable exposition (~200–500 s)	Internal and external acidification abolish channels’ current	Not assessed	Not assessed	[70]
	TRPM7-overexpressing HEK293 cells	Patch clamp	Internal solution with pH 6.1 and ~10 min exposition	Internal acidification decreases TRPM7 currents’ density	Not assessed	Not assessed	[71]
	Mouse hippocampal neurons	Patch clamp	External solution with pH 6.5, 2 min exposition	Extracellular acidification slows down channel’s activation in a voltage-independent way	Not assessed	Not assessed	[72]
	TRPM7-overexpressing HEK293T cells	Patch clamp	External solution with pH 4 and pH 6, acute treatment	External acidification increases channel’s current amplitude in a pH_e_-dependent manner	Not assessed	Not assessed	[69]
	TRPM7-overexpressing HEK293T cells	Patch clamp	External solution with pH 3–7, ~50 s-long exposition	External acidification determines a significant increase in TRPM7 inward current in an [H^+^] in a concentration-dependent manner	Not assessed	Not assessed	[73]
	Pig isolated ventricular myocytes	Patch clamp	External solution with pH 5.5 and pH 6.5, ~5–10 min exposition	External acidification decreases channel’s current amplitude in a pH_e_-dependent and voltage-independent manner. The inhibitory effect of acidic pH_e_ is prevented increasing intracellular pH buffering capacity	Not assessed	Not assessed	[68]
	Rat basophilic leukemia cells (RBL)	Patch clamp	External solution with pH 5.5, pH 6 and pH 6.5, ~1-min-long exposition	External acidification decreases channel’s current amplitude in a pH_e_-dependent manner	Not assessed	Not assessed	[68]
	HeLa cells	Patch clamp Cell death assays (fluometric analysis of caspase 3/7 activation, electronic sizing of cell volume, and triple staining with Hoechst/acridine orange and propidium iodide assay.	External solution with pH 4 and pH 6, acute treatment for patch clamp experiments, and 1 h-long treatment with acidic pH_e_ (4 and 6) for cell death assays	External acidification increases channel’s current amplitude in a pH_e_-dependent manner	Not assessed	Acidosis promotes HeLa necrotic cell death	[74]
	Human atrial cardiomyocytes	Patch clamp	External solution with pH 4–6, acute treatment	External acidification increases channel’s current amplitude in presence of divalent cations in the extracellular milieu	Not assessed	Not assessed	[75]
**TRPV1**	TRPV1-expressing HEK293 cells	Patch clamp	Acidic solution with pH 5.5 applied intracellularly for ~50 s	Acid treatment does not activate the channel in inside-out patches but potentiates 2-APB-evoked currents from the cytoplasmic side	Not assessed	Not assessed	[76]
	hTRPV1-transfected HEK293t cells	Calcium imaging	External solution with pH 4.3 and pH 6.1, ~4 min-long exposition	Acidic pH_e_ activates TRPV1 channel	pH_e_ 6.1 determines larger Ca^2+^ transients with respect to pH_e_ 4.3 in physiological extracellular Ca^2+^ concentration, while, in presence of low extracellular Ca^2+^ concentration, cells exposed to pH_e_ 6.1 show reduced Ca^2+^ entry respect to pH_e_ 4.3 exposition	Not assessed	[77]
	Defolliculated Xenopus laevis oocytes, TRPV1-expressing HEK293 cells	Patch clamp	Extracellular solution with pH 6.4, cells pre-treated with acid bath solution for 2 min	Acidic pH_e_ potentiates heat-evoked TRPV1 current in oocytes; potentiation of capsaicin and heat-evoked TRPV1 currents in HEK293 cells	Not assessed	Not assessed	[78]
	Primary human adult dermal lymphatic endothelial cell (HDLECs)	Cell viability assay Cell invasion assay in vitro tube formation assay Transwell cell migration assay	24 h long exposition to pH_e_ 6.4, and 6 h long exposition for in vitro tube formation assay	Acidic pH_e_ activates TRPV1 channel	Not assessed	Acidic pH_e_ affects HDLECs morphology, increasing their migration and invasive abilities, proliferation and promoting lymphangiogenesis via acidosis-induced TRPV1 activation	[79]
**TRPV2**	TRPV2-expressing HEK293 cells	Patch clamp	Acute administration of extracellular solution with pH_e_ 5.5 and 6	Extracellular acidosis potentiates the response of TRPV2 to 2-APB (and analogues) from the cytosolic side, while intracellular acidification and low pH_e_ alone are not able to elicit any detectable current	Not assessed	Not assessed	[80]
**TRPV3**	TRPV3-expressing HEK293 cells	Patch clamp, calcium imaging	Acute administration of extracellular solution with pH_e_ 5.5 and 6	Extracellular acidosis potentiates the response of TRPV3 to 2-APB (and analogues) from the cytosolic side. Intracellular acidification activates the channel, eliciting small but detectable currents	Extracellular acidosis increases Ca^2+^ entry following 2-APB stimulation	Not assessed	[80]
	TRPV3-expressing HEK293 cells	Patch clamp Cell death assay (PI staining assay)	Intracellular administration of acidic solution with pH_e_ 5.5 and glycolic acid. Extracellular solution with pH 5.5. Intracellular solution with pH 5.5–7.	Glycolic acid-induced intracellular proton release in presence of acidic solution activates the channel in a reversible way. Extracellular acidification does not activate TRPV3, while intracellular acidification alone activates the channel in a pH-dependent manner	Not assessed	Glycolic acid-induced acidification induces cell toxicity and cell death	[81]
	Human keratinocytes cells (HaCaT)	Patch clamp, cell death assay (PI staining assay)	Intracellular administration of acidic solution with pH_e_ 5.5 and glycolic acid	Glycolic acid-induced intracellular proton release in presence of acidic solution potentiates the channel’s response to 2-APB in a reversible manner	Not assessed	Glycolic acid-induced acidification induces cell toxicity and cell death	[81]
**TRPV4**	Chinese hamster ovary cells	Patch clamp	External solution with pH_e_ 4, 5.5 and 6, acute treatment	Extracellular acidosis activates the channel in a pH_e_-dependent manner	Not assessed	Not assessed	[82]
	mTRPV4-overexpressing primary cultured mouse esophageal epithelial cells	Ca^2+^ imaging	External solution with pH_e_ 5, acute treatment	Not assessed	Extracellular acidic pH decreases Ca^2+^ entry, lowering cytosolic Ca^2+^ concentration	Not assessed	[83]
**TRPV6**	Jurkat cells	Patch clamp	External solution with pH 6, acute treatment	Extracellular acidosis suppresses TRPV6-mediated currents	Extracellular acidic pH reduces Ca^2+^ entry, lowering cytosolic Ca^2+^ concentration	Not assessed	[84]
**TRPA1**	HEK-293t cells expressing hTRPA1, mTRPA1, or rTRPA1	Patch clamp Calcium imaging	Acidic solutions with pH 7.0, 6.4, 6.0, and 5.4, 30 s-long treatment in calcium imaging experiments	Extracellular acidosis activates inward currents via hTRPA1 and potentiates acrolein-evoked currents of hTRPA1 in a pH_e_-dependent and reversible manner, while failing to activate mouse and rodent TRPA1.	Extracellular acidosis increases Ca^2+^ entry in hTRPA1, no effect on mTRPA1 and rTRPA1.	Not assessed	[85]
	DRG neurons derived from TRPV1/TRPA1−/− mice and overexpression hTRPA1	Calcium imaging	Acidic solutions with pH 5, 60 s-long treatment	Not assessed	Acidic pH_e_ induces Ca^2+^ entry	Not assessed	[85]
	Neuroblastoma ND7/23 cells expressing hTRPA1	Patch clamp	Acidic solution with pH 5, acute treatment	Acidic pH_e_ activates hTRPA1	Not assessed	Not assessed	[85]
**TRPC5**	TRPC5-transiently transfected HEK293 cells	Patch Clamp	External acidic solution with pH 4.2, 5.5, 6.5, 7, ~100 s-long treatment	G protein-activated and spontaneous currents are potentiated by extracellular acidic pH by increasing the channel open probability, with a maximum effect at ~pH 6.5, while more acidic values inhibit the channel	Not assessed	Not assessed	[86]
**TRPC4**	TRPC4-transiently transfected HEK293 cells	Patch Clamp	External acidic solution with pH 4.2, 5.5, 6.5, 7, ~100 s-long treatment	G protein-activated currents are potentiated by extracellular acidic pH, with a maximum effect at ~pH 6.5 and complete inhibition at pH_e_ 5.5	Not assessed	Not assessed	[86]
	mTRPC4-stably transfected HEK293 cells	Patch Clamp	External acidic solution with pH 6.8	Low pH_i_ (6.75–6.25) accelerates G_i/o_-mediated TRPC4 activation, and this requires elevations in intracellular calcium concentration. Intracellular protons inhibit Englerin A-mediated TRPC4 activation	Not assessed	Not assessed	[87]
**TRPC6**	TRPC6-transiently transfected HEK293 cells	Patch Clamp	External acidic solution with pH 4.2, 5.5, 6.5, 7, ~100 s-long treatment	Acidic pH_e_ inhibits channel’s inward and outward currents starting from pH_e_ 6.5 and the inhibition is potentiated by more acidic pH_e_ values.	Not assessed	Not assessed	[86]
**ORAI1/STIM1**	Human macrophages	Patch clamp	External acidic solution with pH 6 and 8, ~200 s-long treatment	Extracellular acidosis inhibits ORAI1 channel in a pH_e_-dependent and reversible manner	Not assessed	Not assessed	[88]
	H4IIE rat liver cells overexpressing ORAI1 and STIM1	Patch clamp	External acidic solutions with pH 5.1 and 5.9	ORAI1 and STIM1-mediated I_CRAC_ are inhibited by acidic pH_e_, with maximal effect at pH_e_ 5.5	Not assessed	Not assessed	[89]
	RBL2H3 mast cell line, Jurkat T lymphocytes and heterologous ORAI1-2–3/STIM expressing HEK293 cells	Patch clamp	External and intracellular acidic solutions with pH 6 and 6.6	External and internal acidification inhibits IP3-induced I_CRAC_ in RBL2H3 mast cell line, Jurkat T lymphocytes, and in heterologous ORAI/STIM-mediated I_CRAC_ in HEK293 cells in a reversible manner	Not assessed	Not assessed	[90]
	ORAI1/STIM1-transiently transfected HEK293 cells	Patch Clamp	External acidic solution with pH 5.5	Acidic pH_e_ inhibits ORAI1-2–3/STIM1 current amplitude in a reversible and pH-dependent manner, with a maximal effect at pH_e_ 4.5	Not assessed	Not assessed	[91]
	ORAI1/STIM1-transiently transfected HEK293 cells	Patch Clamp	Intracellular acidic solution with pH 6.3	Intracellular acidosis inhibits ORAI1/STIM1 current, regulating the amplitude of the current and the Ca^2+^-dependent gating of the CRAC channels	Not assessed	Not assessed	[92]
	SH-SY5Y human neuroblastoma cells	Ca^2+^ signals quantification by Mn^2+^ quench technique	External acidic solution with pH 6.8 and 7 and 7.2. Different treatment time, ranging from ~3–4 min to ~8 min for carbachol-mediated Ca^2+^ entry and ~7 min for thapsigargin-mediated Ca^2+^ entry	Not assessed	Tumour acidic pH_e_ inhibits carbachol- and thapsigargin-mediated Ca^2+^ entry in a reversible manner, while intracellular acidification or alkalinization leads to no effects in carbachol-mediated Ca^2+^ entry	Not assessed	[93]

**Table 2 ijms-23-07377-t002:** Ca^2+^-permeable ion channels regulation by hypoxia.

Ion Channel	Cell Type	Methodology	Hypoxia Technique and Treatment Time	Effect of Hypoxia on Channel’s Activity/Expression	Effect of Hypoxia on Ca^2+^ Signals	Cellular Function	Ref.
**Piezo1**	Mouse and human sickle red blood cells (RBCs)	Cell-attached and nystatin-permeabilized patch clamp Calcium imaging	Deoxygenation obtained by exposure with a superfusate gassed 30 min prior to the experiment with 100% N_2_	Deoxygenation activates a Ca^2+^- and cation-permeable conductance in a reversible manner, and this current is sensitive to inhibition by GsMTx-4; 1 mM	Increased Ca^2+^ influx	Not assessed	[94]
	Pulmonary arterial endothelial cells (PASMCs) of patients with pulmonary arterial hypertension (PAH)	Calcium imaging EdU and cell counting proliferation assay Western Blot	/	Piezo1 expression and activity are increased in idiopathic pulmonary arterial hypertension and pulmonary arterial smooth muscle cells	Increased Ca^2+^ influx and increased intracellular Ca^2+^ release	Increased PAH-PASMCs’ proliferation	[95]
	Pulmonary artery smooth muscle cells of mice and rats’ models with experimental chronic hypoxia-induced pulmonary hypertension (PH) Human pulmonary artery endothelial cells (hPAECs)	Western Blot Calcium imaging	Hypoxia induced by incubation in 3% O_2_ for 4 h–12 h or in 10% O_2_ for a total of 6 weeks	Piezo1 is significantly upregulated in the lung tissue of PH rats and in chronic hypoxia-induced PH models. Piezo1 protein is transiently upregulated also in hPAECs after 6 h exposition to hypoxic conditions. Hypo-osmotic conditions upregulate Piezo1 protein levels in hPAECs	Hypo-osmotic upregulation of Piezo1 promotes Ca^2+^ influx, promoting Akt and Erk signalling pathways activation, with downstream upregulation of Notch ligand	GsMTx4-mediated Piezo1 blockade partially reduces the chronic hypoxia-induced PH in mice with chronic hypoxia-induced pulmonary hypertension	[96]
**TRPM2**	TRPM2 WT and knockout (KO) neonatal hypoxic-ischemic (HI) brain injury mouse model	Western Blot	Hypoxia damage was induced in ischemic mice models by incubating the pups in a hypoxic chamber for 2 h	TRPM2 is acutely overexpressed 24 h after hypoxia-ischemic injury in brain tissue samples from mouse pups	Not assessed	Brain damage and inflammation are reduced in TRPM2 KO mice 7 days following hypoxic-ischemic brain injury. TRPM2 inhibits cell survival pathways after HI injury	[97]
	Primary cultures of rat cortical neurons subjected to oxidative stress	Calcium imaging Trypan Blue exclusion assay	Oxidative stress induced by 1 mM or 50 µM H_2_O_2_ treatment	Not assessed	H_2_O_2_ induces TRPM2-mediated intracellular calcium rise	SiTRPM2 prevents H_2_O_2_-mediated neuronal cell death	[98]
	TRPM2-overexpressing HEK293 cells	Whole-cell Patch Clamp	Hypoxia induced by cell incubation with gas mixture containing 5% O_2_ for 30 and 60 min	TRPM2 activation is induced by 30- and 60-min exposure to hypoxic conditions	Not assessed	Hypoxia treatment enhances cell death, probably via TRPM2-mediated Ca^2+^ influx	[99]
	ARPE-19 retinal pigment epithelial cells	Patch Clamp Calcium imaging Propidium iodide cell death assay	Hypoxia induced by CoCl_2_ (200 μM) for 24 h	Hypoxia induces activation of TRPM2 currents and upregulates TRPM2 protein levels	Hypoxia induces TRPM2-mediated intracellular calcium rise	Hypoxia causes mitochondrial oxidative cell cytotoxicity and cell death via TRPM2-mediated Ca^2+^ signals	[100]
	Primary IGR39 melanoma cells TRPM2-overexpressing HEK293 cells	Patch Clamp Calcium imaging	Treatment with chloramine-T (Chl-T) oxidant agent	Amount of 0.5 mM Chl-T activates TRPM2 in IGR39 and in TRPM2-expressing HEK293 cells	Chl-T treatments induce a significant increase in cytosolic Ca^2+^ levels	Chl-T-induced TRPM2 activation and increased Ca^2+^ influx activate BK and K_Ca_3.1 potassium channels	[101]
	PC3 prostate cancer cells	Calcium imaging MTT and TUNEL assay	Treatment with 0.5 to 4 mM H_2_O_2_ for 6 h	H_2_O_2_ induces TRPM2 activation	H_2_O_2_ treatment leads to TRPM2-mediated intracellular Ca^2+^ increase in a concentration-dependent manner	H_2_O_2_ induces TRPM2-Ca^2+^-CaMKII cascade that promotes ROS production, mitochondrial fragmentation, and inhibition of autophagy, inducing cell death	[102]
	TRPM2-L and TRPM2-S-expressing SH-SY5Y neuroblastoma cells	Calcium imaging	Treatment with 250 μM H_2_O_2_ for 20 min	Not assessed	H_2_O_2_ treatment leads to TRPM2-L-mediated intracellular Ca^2+^ increase and a decrease in TRPM2-S	TRPM2-L-expressing cells show higher HIF-1/2α levels with respect to TRPM2 short isoform and promote tumour growth in vivo	[103]
	Human breast cancer cells	Calcium imaging qPCR	Co-culture with neutrophils or H_2_O_2_ treatment	Neutrophil-derived H_2_O_2_ induces decrease in TRPM2 expression in H_2_O_2_-selected tumour cells	Not assessed	TRPM2 activation by neutrophil-derived H_2_O_2_ and following Ca^2+^ entry promotes cancer cells’ death	[104]
**TRPM6**	Hepatic ischemia-reperfusion rat model	qPCR	Ischemia was obtained by 60 min clamping the left hepatic artery and the portal vein	TRPM6 expression is increased in liver tissue from ischemia-reperfusion rat model	Not assessed	Not assessed	[105]
**TRPM7**	TRPM7-overexpressing HEK293T cells Cortical neurons	Ca^2+^ imaging Patch clamp PI cell death assay	Hypoxia induced by anaerobic chamber containing ˂0.2% O_2_ atmosphere for 1, 1.5 and 2 h.	Hypoxia induces TRPM7 channel activation	Hypoxia increases Ca_2+_ entry	Hypoxia-activated TRPM7 mediated-Ca^2+^ entry determines cell death in cortical neurons	[106]
	Hepatic ischemia-reperfusion rat model	qPCR	Ischemia was obtained by 60 min clamping the left hepatic artery and the portal vein	TRPM7 expression is increased in liver tissue from ischemia-reperfusion rat model	Not assessed	Not assessed	[105]
**TRPV1**	HEK293T cells overexpressing rat TRPV1	Patch Clamp Calcium imaging	Hypoxic solution obtained by bubbling with 100% N2 gas for at least 20 min before the perfusion (PO_2_, 3%)	Acute hypoxia weakly increases TRPV1 activity, but negatively affects capsaicin induced TRPV1 currents	Hypoxia leads to a slight increase in cytosolic Ca^2+^ levels	Not assessed	[107]
	Rat DRG neurons hTRPV1/rTRPV1-expressing HEK293 cells	Whole-cell patch-clamp	Overnight (18–20 h) exposition to hypoxia (4% O_2_)	Overnight exposure to hypoxic/high glucose conditions increases TRPV1 mean peak current densities in both cell lines, without affecting its expression	Not assessed	Not assessed	[108]
	Rat pulmonary artery smooth muscle cells (PASMCs)	Calcium imaging qPCR Western Blot Wound Healing assay BrdU proliferation assay	24–48 h long exposition to hypoxia (1% and 10% O_2_)	Hypoxic conditions do not affect TRPV1 expression, but they increase TRPV1 activity	No assessed	Hypoxia-mediated TRPV1 activation enhances PASMCs migratory abilities and proliferation	[109]
	Human pulmonary artery smooth muscle cells (PASMCs)	Calcium imaging qPCR Western Blot Cell count proliferation assay	72 h long exposition to hypoxia (3% O_2_)	Chronic hypoxia upregulates both TRPV1 gene and protein levels	Chronic hypoxia increases cytosolic Ca^2+^ levels	The proliferation of PASMCs is increased under hypoxia	[110]
**TRPV2**	HepG2 and Huh-7 human hepatoma cell lines	RT-PCR Western Blot Flow cytometry	50, 100, 200, and 400 Μm H_2_O_2_ treatment for 24 h	H_2_O_2_ upregulates the expression of TRPV2 at mRNA and protein levels	Not assessed	Overexpression of TRPV2 promotes H_2_O_2_-induced cell death	[111]
**TRPV3**	Rat myocardial cells	MTT and Edu staining assay Western Blot Caspase-3 and LDH activity assay	12 h long exposition to hypoxia (1% O_2_)	TRPV3 is overexpressed in myocardial cells induced by ischemia/hypoxia	Not assessed	TRPV3 silencing protects cardiomyocytes from hypoxia-induced cell death and decreases the secretion of proinflammatory cytokines	[112]
	Primary rat pulmonary artery smooth muscle cells (PASMCs)	Western Blot Flow cytometry MTT assay	24 h long exposition to hypoxia (3% O_2_)	TRPV3 protein expression is enhanced in PASMCs from hypoxic rats	Not assessed	TRPV3 mediates hypoxia-induced PASMCs’ proliferation via PI3K/AKT signalling	[113]
	TRPV3-overexpressing HEK293	Patch Clamp	12 h long exposition to hypoxia (1% O_2_)	Pre-incubation in hypoxic conditions potentiates TRPV3 currents in response to 2-APB treatment	Not assessed	Not assessed	[114]
**TRPV4**	Rat cardiomyocytes	Western Blot qPCR Calcium imaging	6 h long exposition to hypoxia (95% N_2_) in a controlled hypoxic chamber	TRPV4 gene and protein expression levels are increased after 6 h exposure to hypoxia	Hypoxia increases TRPV4-mediated Ca_2+_ influx responses to 300 nM GSK	Hypoxia-mediated activation of TRPV4 induces cytosolic Ca^2+^ overload in cardiomyocytes, leading to ROS production and oxidative injury in vitro and in vivo	[115]
	Adult rat hippocampal astrocytes	Patch Clamp qPCR Western Blot Calcium imaging	Hypoxia/ischemia (H/I) is induced by occlusion of the common carotids in combination with hypoxic conditions (from 1 h up to 7 days, 6% O_2_)	TRPV4 mRNA and protein expression are significantly increased 1 h after H/I. H/I also activates TRPV4 channel	H/I enhances the response of 4aPDD, inducing TRPV4-mediated Ca^2+^ oscillations	Not assessed	[116]
**TRPA1**	Several breast and lung cancer cell lines	Calcium imaging Cell viability and apoptosis assay via PI and Annexin IV staining	Treatment with 10 µM H_2_O_2_ for 15 min for calcium measurements, 1, 20, and 100 µM for 72–96 h-long exposition for cell viability and cell death assays	H_2_O_2_ treatment activates TRPA1 channel	H_2_O_2_ treatment increases TRPA1-mediated calcium entry	TRPA1-mediated calcium entry promotes cell survival by upregulating anti-apoptotic pathways and promoting oxidative stress resistance	[117]
	Oligodendrocytes	Calcium imaging	Ischemia inducing solution	Not assessed	Ischemia-induced intracellular acidosis promotes Ca^2+^ entry via TRPA1	Ischemia-induced intracellular acidosis and consequent Ca^2+^ entry via TRPA1 mediate myelin damage	[118]
**TRPC1**	U-87 MG glioma cells	qPCR, western blot	Hypoxia induced by exposition to 1% O_2_	Not assessed	Not assessed	TRPC1 participates in hypoxia-induced VEGF gene and protein expression	[119]
	MDA-MB-468 breast cancer cells	qPCR, calcium imaging	Hypoxia induced by exposition to 1% O_2_ for 24 h	Hypoxia upregulates TRPC1 via HIF1α	siTRPC1 reduces non-stimulated Ca^2+^ entry and increases Store-Operated Ca^2+^ entry in hypoxic conditions	TRPC1 overexpression promotes Snail EMT marker upregulation and decrease in claudin-4 epithelial marker in hypoxic conditions. TRPC1 regulates HIF-1α protein levels via Akt-dependent pathway and promotes hypoxia-induced STAT3 and EGFR phosphorylation. TRPC1 also regulates hypoxia-induced LC3BII levels via effects on EGFR.	[120]
**TRPC5**	MCF-7/WT and adriamycin-treated (MCF-7/ADM) human breast cancer cells	Western Blot, immunofluorescence,	Not assessed	Not assessed	Not assessed	TRPC5 promotes HIF-1alpha translocation to the nucleus and HIF-1alpha-mediated VEGF expression, boosting tumour angiogenesis	[121]
	SW620 colon cancer cells	Western blot, transwell invasion, and migration assay, MTT proliferation assay	Not assessed	Not assessed	Not assessed	TRPC5 activates HIF-1alpha-Twist signalling to induce EMT, supporting colon cancer cells’ migration, invasion, and proliferation	[122]
**TRPC6**	Murine pancreatic stellate cells (mPSCs)	Time-lapse single-cell random migration assay Bead-based cytokine assay qPCR Western Blot Ca^2+^ signals quantification by Mn^2+^ quench technique	24 h incubation in hypoxic conditions (1% O_2_, 5%CO_2_, and 94% N_2_) or chemically induced hypoxia by pretreatment with 0.5 mmol/l DMOG	Hypoxic conditions enhance TRPC6 expression and activate the channel	Hypoxia stimulates Ca^2+^ influx mediated by TRPC6 channels	Hypoxia-induced TRPC6 activation enhances mPSCs migration via secretion of pro-migratory factors	[123]
	lx-2 human hepatic stellate cells (HSCs)	Calcium imaging qPCR Western Blot	Hypoxia induced by 100 μmol/L CoCl_2_ treatment	Hypoxic conditions enhance TRPC6 expression and activate the channel	Hypoxia stimulates Ca^2+^ influx mediated by TRPC6 channels	Hypoxia-induced TRPC6 activation and consequent calcium entry promote the synthesis of ECM proteins, which facilitate the fibrotic activation of HSCs	[124]
	Huh7 and HepG2 hepatocellular carcinoma cells (HCCs)	Confocal Calcium imaging Western Blot	Hypoxia induced by cell incubation in a low oxygen atmosphere with 1% O_2_, 5%CO_2_, and 94% N_2_ for 6 h	Hypoxic conditions activate the channel	Hypoxia promotes calcium influx	Hypoxia-induced TRPC6-mediated calcium entry promotes HCCs drug resistance via STAT3 pathway	[125]
	U373MG and HMEC-1 glioblastoma cell lines	qPCR Western Blot Calcium imaging Proliferation assay Matrigel invasion assay Endothelial cell tube formation assay	Hypoxia induced by 100 μmol/L CoCl_2_ treatment	Hypoxia enhances TRPC6 expression via Notch pathway	Hypoxia stimulates Ca^2+^ influx mediated by TRPC6 channels	Hypoxia-induced TRPC6-mediated calcium entry promotes HCCs proliferation, colony formation, and invasion via NFAT pathway	[126]
**ORAI1/STIM1**	Primary Aortic Smooth Muscle Cells and HEK293 cells transfected with ORAI1 and STIM1	Patch Clamp Calcium imaging	Hypoxia was induced with 3 methods: (1) sodium dithionite (Na_2_S_2_O_4_) treatment to 1 mM final concentration, pH adjustment to pH 7.4, and bubbling with 100% N_2_. (2) cell culture media with 30 min-long bubbling with 100% N_2_. (3) cell culture media with 30 min-long bubbling with 3% O_2_	Intracellular acidification induced by hypoxia in HEK293 cells leads to inhibition of SOCE by disrupting the electrostatic ORAI1/STIM1 binding and closing ORAI1 channel.	Hypoxia-induced intracellular acidification reduces SOCE in Primary Aortic Smooth Muscle Cells and HEK293 cells transfected with ORAI1 and STIM1	Not assessed	[92]
	A549 non-small cell lung cancer cells	Western Blot qPCR BrdU cell proliferation assay Calcium imaging Scrape-wound migration assay Matrigel transwell invasion assay	Hypoxia induced by Nicotine treatment for 48 h	Nicotine treatment-induced hypoxia determines ORAI1 overexpression at gene and protein levels	Nicotine treatment-induced hypoxia increases intracellular basal calcium levels and SOCE	Nicotine treatment-induced hypoxia increases A549 cells’ proliferation and migration	[127]
	MDA-MB 231 and BT549 breast cancer cell lines and Human Microvascular Endothelial Cell line-1 (HMEC-1)	Western Blot qPCR Calcium imaging Migration assay (Wound healing and transwell migration assay) Matrigel transwell invasion assay Tube formation assay in vitro	Hypoxia induced by cell incubation in low oxygen atmosphere	Hypoxia promotes ORAI1 gene and protein upregulation via activation of Notch1 signalling	Hypoxia increases thapsigargin-induced SOCE, with consequent rise in cytosolic calcium entry	Hypoxia-induced ORAI1 overexpression and consequent increase in SOCE promote NFAT4 activation and enhance neuroblastoma cells’ migration, invasion, and angiogenesis	[128]
	HCT-116 and SW480 human colon cancer cells and Human Microvascular Endothelial Cell line-1 (HMEC-1)	Western Blot qPCR Calcium imaging Transwell migration assay Matrigel transwell invasion assay Tube formation assay in vitro Cell attachment and detachment assays	Hypoxia induced by 100 μmol/L CoCl_2_ treatment	Hypoxia promotes ORAI1 gene and protein upregulation via activation of Notch1 signalling	Hypoxia increases thapsigargin-induced SOCE	Hypoxia-induced ORAI1 overexpression and consequent increase in SOCE promote NFATc3 activation and enhance neuroblastoma cells’ migration, invasion, and angiogenesis	[129]

### 2.1. Piezo Channels

Piezo proteins are mechanically activated non-selective cations channels identified over a decade ago by Patapoutian’s group [130], a discovery that paved the way for various works that further clarified the role of these channels not only in the transduction of mechanical signals, but also in other physiological processes, and culminating in the 2021 Nobel Prize in Physiology or Medicine for David Julius and Ardem Patapoutian “for their discoveries of receptors for temperature and touch”. Piezo channels play a key role in mechanotransduction, directly responding and integrating mechanical stimuli and forces from the cell environment into biological signals, leading to the activation of Ca^2+^-dependent processes or cell depolarization in different organs [131]. Piezo channels are also involved in other physiological processes, exhaustively reviewed in [131].

The role of Piezo channels in cancer has been deepened in recent years, with several studies evidencing their importance in different types of cancer that originate from tissues subjected to mechanical stress [132]. Piezo1 and/or Piezo2 are overexpressed in several cancers of epithelial origin, where these channels act as oncogenes, enhancing carcinogenesis through different Ca^2+^-dependent signalling pathways [133,134,135,136,137].

In addition to being sensitive to mechanical stimulation, Piezo1 is also regulated by protons. The work of Bae C. and colleagues in 2015 demonstrated that conditions of acidosis (pH_e_ 6.3) inhibit Piezo1 by stabilizing its inactivated state [62]. A drop in extracellular pH and inactivation of Piezo1 might represent a protective mechanism in specific cell types, especially considering that low extracellular pH can promote intracellular acidification and, therefore, the activation of specific signalling pathways that result in cell death [138] (Table 1). This notion is supported by the study of Kuntze A. et al. of 2020, which demonstrated that extracellular acidosis-induced low intracellular pH (pH_i_ 6.7) of pancreatic stellate cells (PSCs) inhibited the activity of Piezo1, reducing the Ca^2+^ influx in PSCs (Figure 1 and Table 1). In these conditions, Piezo1 activation with Yoda1 led to a loss of PSCs spheroid integrity and increased fragmentation, resulting virtually from Ca^2+^ overload and its induction of cell death. Therefore, extracellular acidosis-mediated intracellular pH drop and inactivation of Piezo1 might represent a protective mechanism for PSCs, in which Ca^2+^ fluxes are decreased and apoptosis is avoided [63].

Information about Piezo channels’ regulation by low oxygen pressures is absent in cancer and the literature data are limited to a few works in red blood cells and pulmonary endothelial and smooth muscle cells, where the pathogenic role of Piezo1 was explored.

Stretch-activated cation channel expression, like Piezo1, has been reported in red blood cells (RBCs), where Piezo1 is expressed and localises at the cell membrane, where it regulates RBCs volume. This role assumes major relevance in pathological conditions, such as Sickle cell disease (SCD), where RBCs showed decreased deformability and higher intracellular Ca^2+^ levels with respect to normal red blood cells [139]. In sickle RBCs, low oxygen concentration led to the activation of transmembrane mechanosensitive ion channels. This activation originated a non-selective ion current named P_sickle_, which activated the calcium-activated potassium channel K_Ca_3.1, which initiates the dehydration cascade in sickle RBCs, responsible for potentiating haemoglobin aggregation [140]. Therefore, Ca^2+^ entry in sickle RBCs is necessary for their sickling, which is mediated by the opening of plasma membrane Ca^2+^-permeable and mechanosensitive ion channels. In this context, Piezo1 can be suggested as a mediator of the non-selective ion current, as deoxygenation-induced P_sickle_ is abolished by GsMTx4, the tarantula spider toxin which is an inhibitor of mechanosensitive ion channels, including Piezo1 [94].

Recent works have shown that Piezo1 is upregulated in pulmonary arterial endothelial cells of patients with pulmonary arterial hypertension (PAH) and in pulmonary artery smooth muscle cells of mice and rats models with experimental chronic hypoxia-induced pulmonary hypertension (PH) [95,96]. Human pulmonary arterial endothelial cells showed maximal Piezo1 expression at 6 h hypoxic exposure, with a consequent increase in calpain-1 and calpain-2, both involved in PAH development [96]. Moreover, hypo-osmotic conditions upregulated Piezo1 protein levels in the same cells and promoted the activation of Akt and Erk signalling pathways via Piezo1-induced Ca^2+^ entry, as demonstrated by siPiezo1 treatment, with downstream upregulation of Notch ligands [96]. The potential therapeutic effect of Piezo1 blockade in the mouse model with chronic hypoxia-induced PH was assessed by GsMTx4 treatment, which partially reduced the chronic hypoxia-induced PH [96]. These data support the role of Piezo1 in the remodelling of the pulmonary vasculature in PH.

### 2.2. Transient Receptor Potential Channels

Transient receptor potential (TRP) ion channels are a family of 28 different proteins in humans, mostly permeable to Ca^2+^ ions, characterized by a polymodal activation, and whose altered expression and/or functionality have been linked to several cancer types [54]. Several TRP channels are sensitive to changes in intra- and extracellular pH and to hypoxia (see Table 1 and Table 2), altering Ca^2+^ downstream signalling pathways as described in the following paragraphs.

#### 2.2.1. TRP Melastatin Subfamily

**TRPM2** is a non-selective cation channel localized at the plasma membrane and/or in lysosome compartments and permeable to Ca^2+^, Mg^2+,^ and monovalent cations. TRPM2 is activated by ADP-ribose (ADPR) and by intracellular Ca^2+^ increase associated with oxidative stress and reactive oxygen species (ROS) production [141,142,143,144].

TRPM2’s activity is regulated by both intra- and extracellular pH. The work of Starkus and colleagues demonstrated that extracellular acidic pH (IC_50_ pH = 6.5) inhibited both inward and outward TRPM2 currents in TRPM2-overexpressing HEK293 cells in a voltage-dependent manner, by affecting the single-channel conductance, most probably due to the interaction of protons with outer pore and competing for binding sites with extracellular Ca^2+^ ions, as these ions attenuated the inhibitory effect of pH_e_ on TRPM2 [64]. These results were confirmed the same year by Du J. et al. and on the same cell line, although assuming a non-proton permeation through the channel [65], and by Yang W. et al. in 2010 [66], showing that low pH_e_-mediated inhibition of TRPM2 might be induced by conformational changes following protons binding. Interestingly, extracellular acidic pH effects are species-dependent, with mTRPM2 channels showing less sensitivity to acidic pH_e_ compared to hTRPM2 [67] (see also Figure 1 and Table 1). Du J et al., as well as Starkus et al., also studied the intracellular acidic pH role showing a reversible inhibitory effect on the channel, inducing its closure without affecting single-channel conductance [64], probably by a mechanism of proton competition with the Ca^2+^ and ADPr binding site [65].

Several reports have highlighted the importance of the role of hypoxia in TRPM2’s expression and activity, in particular in the context of hypoxia-induced brain damage, but with little evidence in tumour cells (Figure 1 and Table 2).

Concerning TRPM2’s role in cerebral hypoxia, several comprehensive reviews have elucidated the current knowledge on this topic [145,146]. Huang and colleagues explored the TRPM2 pathological role in neonatal hypoxic-ischemic brain injury mouse model, demonstrating that genetic deletion of this channel had a long-term neuroprotective effect, reducing brain damage and inflammatory responses 7 days following the hypoxia-ischemic injury with respect to wild-type littermates. At the molecular level, TRPM2 was acutely overexpressed 24 h following hypoxia-ischemic injury in brain tissue samples from CD1 mouse pups and it inhibited pro-survival pathways in control mice by decreasing pAkt and pGSK-3β levels, exerting its protective role in TRPM2-null mice via Akt/Gsk-3β pathway [97]. In vitro TRPM2 silencing had a neuroprotective effect also in primary cultures of rat cortical neurons subjected to oxidative stress following brief H_2_O_2_ exposure, reducing the intracellular calcium levels and preventing H_2_O_2_-mediated neuronal cell death [98]. In addition to neuronal cells, other works have explored hypoxia-induced activation of TRPM2 in other cell types, such as HEK-293 cells, where TRPM2 was activated after 30 and 60 min exposure to hypoxic conditions, leading potentially to an increased Ca^2+^ influx and oxidative stress, resulting in cell death [99], and retinal pigment epithelial cells, where 24 h-long exposure to hypoxia promoted ROS production and cell death via TRPM2-mediated Ca^2+^ influx [100].

In the context of tumourigenesis, TRPM2 is upregulated in several cancers where it mediates Ca^2+^-dependent pathways promoting cell survival [147,148,149,150,151,152]. TRPM2 exerts this effect in particular by protecting cancer cells from oxidative stress by increasing their antioxidant defence. Indeed, although high levels of ROS are detected in almost all types of cancer, a precise balancing of their intracellular presence is required to avoid the toxic effect of these reactive species, achieved through the expression of antioxidant proteins. TRM2 channels act as ROS sensors, being activated by oxidative stress and switching on Ca^2+^-dependent signalling pathways that lead to the enhanced activation of transcription factors involved in antioxidants’ expression, (i.e., HIF-1/2a; CREB; NrF2). These factors promote autophagy, DNA integrity, mitochondrial function, and ATP production [147] (Figure 1). Oxidative-stress-mediated TRPM2 activation and consequent rise in intracellular Ca^2+^ levels might promote cancer progression also by activating Ca^2+^-dependent K^+^ channels, such as the large-conductance voltage-dependent BK channel and the medium-conductance voltage-independent K_Ca_3.1 channel, as reported in melanoma cells [101], which role in cancer hallmarks, such as cell viability and cell migration and invasion have been described [59]. However, other studies correlated TRPM2 expression with a higher sensitivity to chemotherapy. Indeed, an anti-survival role was highlighted in breast and colon cancer, where TRPM2 activation by chemotherapy agents resulted in Ca^2+^ entry, intracellular Ca^2+^ overload, and increased mitochondrial depolarisation, leading to cell death [153], and in prostate cancer, where H_2_O_2_-induced TRPM2 activation resulted in PC3 cells’ death via Ca^2+^-dependent inhibition of autophagy [102] (Figure 1). These results could be explained considering the differential role of full-length TRPM2 and its dominant-negative short isoform [103]. Chen and colleagues elucidated their role both in in vitro and in vivo, demonstrating that full-length TRPM2 (TRPM2-L) had a protective role from oxidative stress by increasing antioxidant enzymes and pro-survival transcription factors, while the expression of the dominant-negative short isoform (TRPM2-S) reduced intracellular calcium influx in response to H_2_O_2_ treatment and cell survival [103].

Another important consideration is the key role of TRPM2 in neutrophil-mediated cytotoxicity. Neutrophils secrete H_2_O_2_, which activates TRPM2 expressed on cancer cells’ surfaces. This activation leads to Ca^2+^ influx in cancer cells, resulting in intracellular overload and induction of cell death, as was demonstrated in breast cancer cells [104] (Figure 1). Therefore, despite the pro-proliferative role of TRPM2 in several cancer cell lines, inhibition of its activity by tumour extracellular acidic pH might result in cancer cells’ protection from neutrophil cytotoxicity, with overall major efficiency in dissemination. Finally, TRPM2 is also localised in lysosomal membranes, where the highly acidic pH inside the compartment might prevent TRPM2 activation and Ca^2+^ release from lysosomes to the cytosol, which may trigger apoptosis [154] (Figure 1). These works reinforce the concept that TRPM2 inhibition by acidic pH_e_ and activation by hypoxic conditions represent a protective mechanism for cancer cells.

**TRPM6** and **TRPM7** are both permeants mainly to Mg^2+^ ions and contribute to its homeostasis. These channels are also permeant to Ca^2+^ ions, increasing their intracellular concentration. TRPM7 has a unique structure as a “chanzyme” due to the presence of a kinase domain in its structure. TRPM6 has a tissue-specific expression and it is downregulated in several cancer types, while TRPM7 is ubiquitously expressed and mostly upregulated in different malignancies, where it plays a key role in promoting different cancer hallmarks [155].

TRPM6 and TRPM7 channels’ activity is modulated by both intra- and extracellular pH. For what concerns the effects of intracellular pH on TRPM7, the outward current density was decreased at low intracellular pH in HEK293 cells, with an IC_50_ of pH 6.32 and in the absence of Mg^2+^ [71]. This result was previously obtained also in RBL cells, where native TRPM7 currents were inhibited by intracellular acidification obtained by acetate treatment, and in TRPM7-overexpressing HEK293 cells in the same work [70]. Intracellular alkalinisation, induced by NH_4_^+^ extracellular application, determined the induction of native TRPM7 current and the enhancement of its activity in RBL cells [70,71] (Table 1).

The effects of acidic pH_e_ (<6.0) on TRPM7 activity are quite controversial, with some evidence showing TRPM7 currents inhibition by low pH_e_ [68,70,71,72], others potentiation of TRPM7 inward current by strong acidic pH_e_ in HEK293 cells [69,73] and HeLa human cervical cells [74] (Table 1). The discrepancy of pH_e_ modulation on TRPM7 described by these works could be explained by taking into consideration the importance of Ca^2+^ and Mg^2+^ ions present in the extracellular milieu. According to the work of Mačianskienė R. et al., both TRPM7 outward and inward currents, expressed by cardiomyocytes, were potentiated in the acidic extracellular medium (pH < 6) containing divalent Ca^2+^ and Mg^2+^ ions, while the absence of these ions in the acidic extracellular medium resulted in the low pH_e_-mediated inhibition of TRPM7 currents in a voltage-independent manner [75]. It is hypothesized that protons and divalent cations compete for a binding site within the channel pore, and the absence of these cations might allow protons to permeate the channel and to bind to specific intracellular inhibitory sites, the same bound by intracellular protons, leading to channel inhibition [73,75].

Very few data are available for TRPM6, where contrasting data are reported, as for TRPM7. TRPM6 was indeed inhibited by extracellular acidity similarly to TRPM7 in isolated pig myocytes [68] but potentiated by extracellular acidic pH in HEK293 cells, even though the magnitude of increase in TRPM7 inward current was higher than that of TRPM6 [69] (Table 1).

Concerning TRPM6 and TRPM7 regulation by hypoxia, their expression is increased in a hepatic ischemia-reperfusion rat model [105], while hypoxia-induced TRPM7 overexpression and increased intracellular Mg^2+^ concentrations were reported in rat hippocampal neurons in vitro [156]. TRPM7 sensitivity to hypoxia was also demonstrated by Tymianski and Mori group, showing that TRPM7 is activated by anoxic conditions or treatment with a hypoxic solution prepared by bubbling N_2_ gas [106,157] (Table 2).

Although evidence of the effect of acidic pH and hypoxia on TRPM6/7 in cancer cells is still lacking, several works have demonstrated that TRPM7 is upregulated in various cancers and it is involved in the enhancement of a variety of cancer-related processes regulated by Ca^2+^ signalling, such as proliferation, migration, invasion, cell death escape, and survival and epithelial–mesenchymal transition (EMT), via the activation of the Ras-ERK and the PI3K/AKT/mTOR signalling pathways [155] (Figure 1). Consequently, its activation by cancer-related extracellular acidic pH highlights its pivotal role in cancer progression.

#### 2.2.2. TRP Vanilloid Subfamily

**TRPV1** is a non-selective cation channel with relatively high permeability to Ca^2+^ ions, which is a major player in pain perception activated by different factors, including heat, inflammation, and acidic environment as revealed by David Julius, that shared the 2021 Nobel Prize for Physiology or Medicine [158,159,160]. Indeed, TRPV1 plays a key role in acidosis-induced pain, acting as a proton channel and being directly activated by protons [78,161].

Acidosis modulates TRPV1 activity, promoting its activation and potentiating its response to 2-APB, heat, and capsaicin (TRPV1 selective agonist) [76,78,80] (Table 1). hTRPV1 is indeed activated by mild acidosis (pH_e_ 6.1) increasing intracellular Ca^2+^ levels, while the channel is blocked in presence of strong acidic conditions [76]. T633 residue in the pore helix and V538 residue in the S3–S4 linker were identified as key residues involved in extracellular pH sensing [77]. Low pH_e_ (<5.9) significantly potentiated heat and capsaicin-evoked response in HEK293 cells by increasing the channel’s open probability at room temperature, therefore lowering the threshold for the channel activation, even in absence of chemical stimuli [78]. These data suggest that the potentiating effect of capsaicin and protons on TRPV1 are independent of each other and they are mediated by different TRPV1 residues (Table 1 and Figure 1). Concerning intracellular acidification, it enhanced TRPV1 currents evoked by 2-APB, without affecting the ones induced by capsaicin [80]. On the other hand, extracellular acidic pH activated TRPV1 and enhanced lymphatic endothelial cells’ proliferative, migratory and invasive abilities via activation of NF-κB transcription factor and consequent upregulation of IL-8, a lymphangiogenic factor, contributing to lymphatic metastasis in tumour acidic microenvironment context [79] (Figure 1). The role of TRPV1 in cancer is dependent on the cancer type. TRPV1 is indeed upregulated in many cancers and it regulates different cancer cell processes, such as proliferation, cell fate, migration, and invasion in a cancer type-specific manner, via the activation of different Ca^2+^-dependent signalling pathways, such as PI3K/AKT, Ras-ERK and JAK/STAT signalling cascades and NF-κB activation [162], acting both as an anti-proliferative and pro-apoptotic factor in melanoma, colorectal, pancreatic and liver cancer, among others, and exerting a pro-tumour role in highly aggressive types of cancer [162] (Figure 1 and Figure 3). These pro- or anti-tumour effects of TRPV1 can be attributed to the different opening states available to the channel in response of different stimuli, exploiting each opening state’s specific properties for the switching on of specific Ca^2+^-dependent signalling pathways in different cancer cell types. Consequently, the pH_e_ regulation of TRPV1 might be considered as cell type, ligand, and context-specific, making it more difficult to identify its potential role as a pharmacological target.

Hypoxic TRPV1 modulation is reported in several publications, although cancer-related studies are lacking. Whole-cell patch clamp studies in HEK293T cells overexpressing rat TRPV1 showed that acute hypoxia weakly increased TRPV1 activity, but negatively affected capsaicin-induced TRPV1 currents. Hypoxia did not affect acidic pH_e_-activated TRPV1 current. These results were confirmed by Ca^2+^ imaging experiments, where hypoxia induced a slight increase in cytosolic Ca^2+^ levels [107]. A stronger hypoxia-mediated TRPV1 activation was demonstrated in native rat sensory neurons and HEK293 cells expressing rat or human TRPV1, where chronic hypoxia (>24 h) did not affect the channel expression but potentiated TRPV1’s activity via protein kinase C (PKC)ε- and HIF-1α-dependent signalling pathways [108]. Parpaite and colleagues also showed on rat pulmonary artery smooth muscle cells (PASMCs) that hypoxic conditions did not affect TRPV1 expression, but they increased TRPV1 activity probably via its translocation to the plasma membrane, enhancing PASMCs’ migratory abilities and proliferation [109]. Hypoxia-mediated changes in TRPV1 expression were instead reported in human PASMCs, where chronic hypoxia upregulated both TRPV1 gene and protein levels, increasing intracellular Ca^2+^ levels and being involved in PASMCs’ proliferation [110] (Table 2).

In addition to TRPV1, also **TRPV2**, **TRPV3,** and **TRPV4** activity is modulated by acidic pH. **TRPV2** is an intracellular-resident non-selective cation channel that translocates to the cell membrane following PI3K activation. Once at the plasma membrane, TRPV2-mediated Ca^2+^ entry regulates different physiological cellular processes. TRPV2 deregulation has been linked to several types of cancer, where its activity supports its progression, in particular via the activation of the PI3K/AKT and the ERK signalling cascades, by escaping cell death and increasing proliferation, cell migration, and invasion [163]. TRPV2 is known to be insensitive to low pH_e_ alone [76,79], however, it has been demonstrated that acidic pH_e_ (6.0 and 5.5) potentiated TRPV2 currents in transiently transfected HEK293 cells that are evoked by 2-APB, increasing the channel’s sensitivity to this ligand from the cytoplasmic side, as proved by inside-out patch configuration [80] (Table 1 and Figure 1).

Wang and colleagues did not report any change in TRPV2 expression following hypoxia exposure in human PASMCs [110], but H_2_O_2_-mediated oxidative stress upregulated TRPV2 gene and protein expression in two different human hepatoma cell lines, with induction or cell death by activating pro-apoptotic proteins and by inhibiting pro-survival ones, and by enhancing the sensitivity of human hepatoma cells to oxidative stress-associated chemicals [111] (Table 2).

**TRPV3** is a non-selective calcium permeant cation channel mostly expressed in brain and skin, where it is involved in chemo-somatosensing. TRPV3 oncogenic activity was demonstrated in lung cancer, where TRPV3 expression was associated with short overall survival and Ca^2+^-mediated increased proliferation via Ca^2+^/calmodulin-dependent kinase II (CaMKII) [164]. As for TRPV2, acidic pH_e_ alone (pH 5.5) was not able to activate the channel in HEK293 cells, but only to potentiate the TRPV3 response to 2-APB and its analogues via acidification of the intracellular milieu, increasing cytosolic Ca^2+^ levels [80] (Table 1 and Figure 1). Moreover, cytosolic protons activated the channel, inducing small but detectable currents via different mechanisms with respect to 2-APB response potentiation, and indicating four residues in the S2–S3 linker to be implicated in the acid intracellular activation of TRPV3 [80]. A more recent study from the same group has elucidated the mechanism of TRPV3 acid intracellular activation and extracellular inhibition. The authors identified Asp641 residue, localized in the selectivity filter, as a critical residue involved in TRPV3 extracellular acidic pH_e_ inhibition. Intracellular acidification protonated E682, E689, and D727 residues in the C-terminal, facilitating the channel’s sensitization [165]. These data were also obtained by a previous work on TRPV3-transfected HEK293 and in HaCaT cells, where it was shown that TRPV3 is directly activated by glycolic acid-induced cytosolic acidification, inducing cell death, while extracellular acidification failed to activate the channel, resulting instead in decreased current amplitude [81] (Table 1 and Figure 1 and Figure 3).

Concerning hypoxic regulation, no data on cancer cells is available. As for TRPV2, no differences in TRPV3 expression were reported in hypoxia exposed-PASMCs, but a recent work demonstrated TRPV3 upregulation in rat myocardial cells in response to ischemia/hypoxia treatment. To confirm the role of TRPV3 as a mediator of hypoxia-induced inflammation and apoptosis in rat myocardial cells, the authors reported thatTRPV3 silencing protected cardiomyocytes from hypoxia-induced cell death and decreased the secretion of pro-inflammatory cytokines [112]. TRPV3 mediated hypoxia responses also in pulmonary artery smooth muscle cells, promoting their proliferation via PI3K/AKT signalling pathway [113], and hypoxia also robustly potentiated this channel current in TRPV3-overexpressing HEK293 cells in response to 2-APB treatment, without affecting the protein levels [114] (Table 2).

**TRPV4** is a heat-activated and mechanosensitive channel deregulated in different cancer cells, acting mostly as a pro-tumour factor, enhancing cancer cells’ migration and metastasis through the activation of AKT and Rho/ROCK1/cofilin cascade, extracellular remodelling, proliferation and angiogenesis via activation of NFAT and PI3K signalling pathway [166,167,168,169,170,171,172], although a tumour-suppressive role has also been reported, especially in tumours expressing high TRPV4 levels [173,174] (Figure 1 and Figure 3).

For what concerns TRPV4 regulation by pH, low extracellular pH (pH_e_ 6) induced opening of transiently expressed TRPV4 in Chinese hamster ovary cells, with a maximal potentiation at pH_e_ 4, as demonstrated by patch clamp current recording in absence of extracellular Ca^2+^ ions [82]. An opposite effect was observed in mouse oesophageal epithelial cells, where Ca^2+^ imaging experiments showed that the Ca^2+^ influx mediated by TRPV4 was abolished at pH_e_ 5 [83] (Table 1). These opposite results might be explained considering the different techniques used by the two research groups and the experimental conditions in general. TRPV4 is a non-selective ion channel, permeable to protons when the extracellular solution is free from other divalent ions, such as Ca^2+^ ions. Since protons may compete with Ca^2+^ ions, a high extracellular proton concentration might lead to a decrease in Ca^2+^ influx but contributes to the gross TRPV4 current [83]. Collectively, considering the evidence outlined above, TRPV4’s role in tumour biology is cancer type-specific and it might emerge as a potential drug target in the context of cancer treatment.

Regarding hypoxic TRPV4 regulation, no studies on cancer cells are present in the literature, but they are limited to other pathological states. In particular, its mRNA and protein expression levels were increased after 6 h long exposure to hypoxia in cardiomyocytes, inducing Ca^2+^ overload and enhancing oxidative injury [115]. TRPV4 plays also an important role in mediating hypoxia-induced pulmonary vasoconstriction (HPV) [175] and in the hypoxia/ischemia injury in the brain [116] (Table 2).

pH also regulates the activity of another component of the TRP vanilloid family, **TRPV6**, a highly Ca^2+^ selective channel (PCa/PNa~100) that is upregulated in different epithelial cancers, such as prostate, pancreatic, breast and ovarian cancer, in particular during early stages of tumour progression [176]. Several studies has revealed the positive effect of TRPV6 activity on tumour progression through the activation of Ca^2+^-dependent signalling pathways [176], promoting cancer proliferation and cell survival in prostate cancer cells by activating the Ca^2+^-dependent NFAT transcription factor [177,178], invasion of breast cancer cells through Ca^2+^/Calmodulin (CaM)-dependent kinases, such as CaMKII [179,180], cell survival, proliferation, and invasion in pancreatic cancer cells [181] and tumour growth in in vivo ovarian adenocarcinoma xenograft mouse model [182] (Figure 1 and Figure 3).

TRPV6 is a polymodal sensor that is regulated by different chemical and physical stimuli, including acidic pH_e_ and hypoxia. Alkaline pH_e_ positively modulated TRPV6’s activity in Jurkat T-cells, where whole-cell patch-clamp experiments showed that solution at pH 8.2 determined the increase in TRPV6 activity and Ca^2+^ entry, while opposite effects were experienced for acidic pH_e_ (pH = 6), which reduced inward TRPV6 currents and Ca^2+^ influx in Jurkat T-cells [84] (Table 1).

Inhibition of TRPV6 by extracellular acidification might be explained considering TRPV6 expression in distinct stages of cancer progression. Tumours can be characterized by a more acidic extracellular microenvironment during late stages [183], phases in which TRPV6 expression is downregulated in some types of cancers, such as colon cancer [184]. However, the lack of further studies focusing on the effect of extracellular acid pH on TRPV6 activity makes a comprehensive understanding of the role of acid pH_e_ on TRPV6 in the context of tumour acidosis difficult.

As for many other TRPV channels, to date, there is extremely little information about TRPV6 modulation by hypoxia, especially in the tumourigenesis context, with a report of placental TRPV6 overexpression in hypoxic rats [185].

#### 2.2.3. TRP Ankyrin Subfamily

**TRPA1** is a non-selective cation channel that functions as a polymodal sensor, and deregulation is observed in several malignancies in a tissue-specific manner [186]. TRPA1 has been described as highly sensitive to O_2_ and oxidants in vagal and sensory neurons. In particular, TRPA1 is activated in hypoxic conditions by the relief of the inhibition of prolyl hydroxylase (PDH) which is O_2_ sensitive [187].

The role of TRPA1 as a ROS sensor has been demonstrated also in cancer cells. The work of Takahashi et al. has demonstrated a key role of TRPA1 in promoting resistance to ROS-producing chemotherapies and oxidative stress tolerance in breast cancer cells via ROS-mediated TRPA1 activation and Ca^2+^-CaM/PYK2 signalling pathway [117] (Table 2 and Figure 1). TRPA1 interaction with fibroblast growth factor receptor 2 (FGFR2) induces the activation of the receptor, promoting cancer cells’ proliferation and invasion, prompting lung adenocarcinoma metastasisation to the brain [188]. TRPA1 activation might also promote prostate cancer progression by triggering prostate cancer stromal cells’ secretion of VEGF [189], a known mitogenic factor involved in the proliferation, migration and invasion of prostate cancer cells [190]. In line with those results, TRPA1 is indeed also an important player in promoting angiogenesis both in physiological retinal development as well as in prostate cancer-derived endothelial cells [191].

In addition to being activated by oxidants, TRPA1 is activated by several distinct exogenous and endogenous compounds [192] and by protons in the extracellular environment, on which regulation is specie dependent. HEK293 cells expressing human and rodent TRPA1 showed a specie-specific activation of the channel, where only hTRPA1 generated a membrane current when exposed to different extracellular acidic environments (pH_e_ 6.4-5.4) reaching the maximal response at acidic pH_e_ 5.4; this specific effect for hTRPA1 was confirmed by Ca^2+^ imaging experiments in HEK293 cells as well as in DRG neurons derived from TRPV1/TRPA1−/− mice and in neuroblastoma ND7/23 cells expressing hTRPA1, where only hTRPA1 induced an increase in Ca^2+^ entry when exposed to acidic pH_e_. This specie-specific activation of TRPA1 is due to valine and serine residues within transmembrane domains 5 and 6 [85] (Table 1).

TRPA1 can also be activated by intracellular acidification, as observed in the context of ischemia-induced acidification of the extracellular microenvironment in mice oligodendrocytes, with following acidification of the intracellular space and activation of TRPA1. This led to an increase in Ca^2+^ influx and damage to myelin [118] (Table 1). Altogether, these results and their role in cancer suggest that TRPA1 activation by acidic pH_e_ and hypoxia may play a significant role in promoting cancer progression, highlighting its potential as a therapeutic target.

#### 2.2.4. TRP Canonical Subfamily

**TRPC1** is a non-selective cation channel, which assembles to form homo- and heteromeric channels with other members of the family, such as TRPC3, TRPC4 and TRPC5. TRPC1 upregulation has been reported in several cancers, such as pancreatic cancer, where it potentiates BxPc3 cells’ migration via Ca^2+^-dependent activation of PKCα [193], and breast cancer, where it exerts a pro-proliferative role in MCF-7 cells by mediating Ca^2+^ influx induced by K_Ca_3.1 activation [194] and via Ca^2+^-dependent ERK1/2 activation [195]. TRPC1 also promotes human glioma cancer cells’ proliferation via Ca^2+^ entry and supports tumour growth in vivo [196], and lung cancer differentiation, by promoting A549 cell proliferation [197]. TRPC1 also participates with ORAI1 channels in the induction of vimentin, a mesenchymal marker for epithelial–mesenchymal transition (EMT) [198] (Figure 2 and Figure 3).

Although TRPC1 is not regulated by protons, several data demonstrated that TRPC1 expression is hypoxia-mediated. The study by Wang B. et al. in 2009 reported that TRPC1 is functionally expressed in U-87 malignant glioma cells under hypoxia, where it promoted the upregulation of VEGF expression, as VEGF mRNA levels were significantly decreased in presence of TRPC1 inhibitor or RNAi in hypoxic conditions [119] (Table 2 and Figure 2). VEGF has a central role in angiogenesis in both physiological and pathological conditions [199] and solid tumours are characterized by a hypoxic microenvironment, in which the lack of oxygen might promote VEGF expression, in order to induce angiogenesis and increase tumour oxygen supply. TRPC1 mediates hypoxia responses also in breast cancer cells, where HIF-1α promoted its upregulation. In MDA-MB-468 breast cancer cells, TRPC1 is involved in the transactivation of the epidermal growth factor receptor (EGFR) during hypoxia, leading to the increase in LC3B autophagy marker [120]. Moreover, hypoxia-induced TRPC1 activation promoted epithelial–mesenchymal transition in the same cells, upregulating the mesenchymal marker snail and downregulating the epithelial marker claudin-4, promoting the hypoxia-induced EMT and, therefore, the aggressive and invasive phenotype of breast cancer cells [120] (Table 2 and Figure 2).

**TRPC5** is a Ca^2+^-activated ion channel that is regulated by components of the tumour microenvironment, such as acidosis and supports hypoxia responses. In long-term adriamycin-treated breast cancer cells, TRPC5-mediated Ca^2+^ influx promoted HIF-1α translocation in the nucleus and therefore the downstream transcription of HIF-1α-regulated VEGF expression, highlighting its contribution to promoting breast cancer angiogenesis [121] (Table 2 and Figure 2). The same research group also validated TRPC5 role in mediating HIF-1α response in tumour progression in colon cancer, where TRPC5 activated the HIF-1α-Twist signalling pathway to promote EMT, migration and proliferation in SW620 colon cancer cells [122].

TRPC5 also acts as a pH_e_ sensor, as its spontaneous activity and G protein-activated currents are potentiated by extracellular acidic pH by increasing the channel open probability in presence of small changes of extracellular pH, with a maximum activity around pH_e_ 6.5 and current inhibition starting from pH 5.5 [86] (Table 1).

Potentiation by acidic pH_e_ and involvement in HIF-1α regulation demonstrates the interest of cancer cells in keeping TRPC5 channels active and over-expressed, in order to promote cancer-specific hallmarks, in particular chemoresistance. Indeed, the role of TRPC5 in promoting chemoresistance in different types of cancer is well known. In breast carcinoma cells, adriamycin-induced TRPC5 upregulation protects them from chemotherapy treatment, by inducing autophagy via an increase in cytosolic Ca^2+^ concentrations and activation of the Ca^2+^-dependent CaMKKβ/AMPKα/mTOR pathway, promoting the cancer cell survival and tumour growth in vivo [200] (Figure 2). The work by Ma X. et al. in 2012 has also demonstrated in vitro and in vivo the critical role of this channel in promoting chemoresistance of adriamycin-resistant MCF-7 breast cancer cell line with the upregulation of another pump linked to drug resistance. TRPC5 overexpression resulted indeed in Ca^2+^ influx and activation of P-glycoprotein overproduction, a pump in charge of removing cytotoxic drugs from cells via Ca^2+^/calmodulin/calcineurin-dependent NFATc3 signalling pathway [201]. The role of TRPC5 in therapy resistance is not confined to breast cancer, as its upregulation has been also identified in 5-fluorouracil resistant human colorectal cancer cells, where TRPC5 overexpression determines the overproduction of ATP-binding cassette subfamily B member 1 (ABCB1), a pump involved in drug resistance through the export of cytotoxic drugs, via Ca^2+^ entry and activation of Ca^2+^-dependent Wnt/β-catenin signalling pathway and in a glycolysis-dependent manner [201,202,203] (Figure 2).

In addition to chemoresistance, TRPC5 expression was observed to be positively correlated with high proliferative, migratory, and invasive abilities of colon cancer cells, promoting the EMT through the HIF-1α-Twist signalling pathway [122]. Therefore, these results demonstrate that the impact of the acid and hypoxic cancer microenvironment on the TRPC5 channel is aimed at its activation, thus promoting tumour progression (Figure 3).

Similar behaviour is shown by **TRPC4**, which shares high sequence similarity with TRPC5. TRPC4 is a Ca^2+^- and G-coupled receptors-activated non-selective Ca^2+^ permeable cation channel [204]. TRPC4’s role in cancer has been elucidated in the last years, being involved in promoting angiogenesis via cytosolic Na^+^ and Ca^2+^ rise [205,206] and proliferation [207], although a negative effect on A-498 renal cell carcinoma cells via Englerin A-mediated channel activation has been documented [208].

TRPC4-mediated currents are two-fold potentiated when exposed to pH_e_ 6.5; however, lower pH leads to current inhibition starting from pH_e_ 6, with complete current inhibition at pH_e_ 5.5 [86]. G_i/o_-mediated TRPC4 activation is also accelerated by intracellular protons in an indirect way, regulating the kinetics of G_i/o_-dependent TRPC4 activation, and it requires an increase in intracellular Ca^2+^ concentration. Intracellular protons do not act directly on the channel, as they inhibit TRPC4 activation by its direct agonist, Englerin A, but by acting on PLCδ1 [87] (Table 1; Figure 2 and Figure 3). No data are available to our knowledge about hypoxia regulation of TRPC4 expression and/or activity.

**TRPC6** is another TRPC member to be regulated by tumour microenvironmental clues, such as hypoxia. In fact, hypoxic conditions enhanced TRPC6 expression in murine pancreatic stellate cells, which constitute the major cellular components in pancreatic ductal adenocarcinoma’s stroma, and which play a key role in generating PDAC’s characteristic desmoplasia. TRPC6 promoted their activation and it was involved in the secretion of pro-migratory factors in presence of hypoxia [123]. Hypoxia upregulated TRPC6 mRNA expression in hepatocellular carcinoma cells, where TRPC6-mediated Ca^2+^ influx conferred drug resistance to these cells via the Ca^2+^-dependent STAT3 signalling pathway in hypoxic conditions [125]. Hypoxia activated Notch1 and downstream TRPC6 expression also in glioma cells, with a consequent rise in cytosolic Ca^2+^ concentration and Ca^2+^-dependent activation of the calcineurin-NFAT signalling pathway, promoting proliferation, cell invasion and angiogenesis under hypoxia [126] (Table 2, Figure 1 and Figure 3). Hypoxia also upregulated TRPC6 in hepatic stellate cells via HIF-1α/Notch1 pathway, leading to TRPC6-mediated Ca^2+^ influx and the downstream activation of Ca^2+^-dependent nuclear factor of activated T-cells (NFAT) transcription factor and SMAD2/3-dependent TGF-β signalling, which activation resulted in the expression of ECM proteins, such as collagen type I, that facilitate hepatic stellate cells’ fibrotic activation and promotes hepatic fibrosis, strongly linked to arise of hepatocellular carcinoma [124] (Table 2; Figure 1 and Figure 3).

Regarding the direct role of acidic extracellular pH (around pH 6.5), it is sufficient to inhibit TRPC6, and the inhibition increased in a pH-dependent manner, affecting both inward and outwards currents in HEK-transfected cells [86] (Table 1). This inhibitory effect of acidic pH_e_ might be explained considering the high TRPC6 levels expressed by pro-tumourigenic immune cells, such as neutrophils, where TRPC6-mediated calcium entry is required for CXCR2-mediated intermediary chemotaxis [209]. Consequently, inhibition of TRPC6 by acidic pH_e_ may impair neutrophils’ migration and prevent them from leaving the acidic tumour microenvironment, thus contributing to its progression and metastasis by releasing ROS, secreting pro-tumour factors and inducing drug resistance [210]. Altogether, these results suggest that the acidic pH_e_-mediated potentiation of TRPC channels might be restricted to only some members, and it depends on the cell type expressing the channels.

### 2.3. Store-Operated Ca^2+^ Channels

**ORAIs** are Ca^2+^ release-activated Ca^2+^ channels (CRAC) which are major players, with **STIM** proteins, in the mechanism known as store-operated Ca^2+^ entry (SOCE), which mediates Ca^2+^ entry into cells promoting the refilling of ER calcium stores as well as intracellular signalling, controlling both physiological and pathological processes such as inflammation, cell motility, cell proliferation, gene expression, apoptosis escape and cell invasion [211,212,213]. SOCE represents the main route of Ca^2+^ entry in different types of cancer cells, contributing to several cancer hallmarks. Indeed, different works have highlighted the key role of SOCE in promoting the migration of different cancer cell lines, such as chemoresistant IGROV1 ovarian cancer cells by regulating focal adhesion turnover [214], SW480 colorectal carcinoma cells [215] and oral cancer cells through Akt/mTOR/NF-κB signalling [216]. Moreover, SOCE has been implicated in enhancing invasion of triple-negative breast cancer cells, as well as angiogenesis and migration, through NFAT4 signalling [128] and through NFATc3 in colorectal cancer cells and tissues from patients [129], while in WM793 cells melanoma cells, SOCE-induced Ca^2+^ oscillations contribute to invadopodia formation via Src activation [217]. SOCE might promote invasion of cancer cells by inducing epithelial–mesenchymal transition, as observed in DU145 and PC3 prostate cancer cells [218] and BGC-803 and MKN-45 gastric cancer cells [219]. Store-operated channels (SOCs) in general play a key role also in the modulation of sensitivity to chemotherapy in a cancer type-specific manner, by promoting chemoresistance in breast cancer cells in the case of ORAI3 [220], ORAI1 and STIM1 in pancreatic ductal adenocarcinoma [212], ORAI1 in hepatocarcinoma [221], among others. SOCE also promotes extracellular vesicle formation, which is a signalling vector involved in the intercellular acquisition of multidrug resistance, in both malignant and non-malignant breast cancer cells via activation of calpain [222].

In addition to all these contributions to cancer progression, SOCE is regulated by hypoxia and pH. Several studies have demonstrated that intracellular and extracellular pH is able to modulate the activity of ORAI channels by affecting its coupling with STIM1 and/or by modifying its gating biophysical properties. The results obtained by numerous studies conducted on ORAI/STIM have clarified the concept that both intra- and extracellular acidic pH have an inhibitory effect on the activity of ORAIs, and on SOCE in general, while intra- and extracellular basic pH potentiate them. The notion that extracellular pH regulates native CRAC currents (I_CRAC_) was already known in 1995 when the work of Malayev A. and Nelson D.J. showed that acidic extracellular pH (pH_e_ = 6) decreased the amplitude of inward Ca^2+^ currents while basic pH_e_ (pH_e_ = 8) increased it in macrophages by using the patch clamp technique and that these changes were reversible and voltage-independent [88] (Table 1). More recently, inhibition by acidic pH_e_ was demonstrated in H4IIE rat liver cells overexpressing ORAI1 and STIM1 proteins, in which I_CRAC_ was inhibited completely at pH_e_ 5.5 [89] (Table 1). In the same work, researchers identified E106, located in ORAI1’s pore, as the residue responsible for pH dependence of CRAC currents, as E106D mutation in ORAI1 abolished the inhibition of I_CRAC_ by acidic pH_e_. These results were also supported by the work of Beck A. and colleagues, which demonstrated that extracellular and intracellular acidification decreased the amplitude of IP3-induced endogenous I_CRAC_ in RBL2H3 mast cell line, in Jurkat T lymphocytes and in heterologous ORAI/STIM-mediated I_CRAC_ in HEK293 cells [90] (Table 1). In contrast to acidification, external alkalinisation increased both endogenous and overexpressed ORAI/STIM amplitude of I_CRAC_ (pKa of 7.8 for RBL2H3 mast cells, 8.0 for Jurkat T lymphocytes and 7.9 for HEK293 cells). Other two key residues (D110 and D112) located in ORAI1’s first extracellular loop have been proposed to contribute to some extent to pH_e_ sensitivity. Indeed, mutations of these residues to alanine prevented the alkalinisation-induced potentiation of I_CRAC_ and increased its amplitude in the presence of acidic pH_e_ [90]. Enhancement and the decrease in SOCs activity by external basic and acidic pH, respectively, were further confirmed in heterologous ORAI/STIM-mediated currents in HEK293 cells by Tsujikawa H. et al. in 2015, who have also demonstrated that E106 mediates pH_e_ sensitivity when Ca^2+^ is the permeant cation, while E190 when Na^+^ is the permeant cation [91] (Table 1). However, the effect of alkaline pH_i_ on I_CRAC_ is controversial. Indeed, alkaline pH_i_-mediated potentiation of ORAI1/STIM1 activity was observed in other papers [91,223] (Table 1). These differences might be explained considering the type of intracellular Ca^2+^ buffer used [223]. Moreover, cytosolic alkalinisation led to SERCA inhibition, resulting in Ca^2+^ release from ER stores and activation of SOCs, with Ca^2+^ influx in NIH 3T3 cells [224]. Residual H155 located in the intracellular loop of ORAI1 is responsible for ORAI1/STIM1 pH_i_ sensitivity, as mutation to phenylalanine decreased low pH_i_-mediated I_CRAC_ inhibition and alkaline pH_i_-mediated I_CRAC_ potentiation [91]. Since the effect of pH was the same in presence of all ORAI isoforms [90] and that both extracellular pH sensors (residues E106 and E190) and intracellular one (residue H155) are conserved in all three ORAI isoforms, it might suggest that the residues mentioned before act as common pH_i_ and pH_e_ sensors in ORAI1–2–3/STIM isoforms. In addition to H155, negatively charged amino acid residues in the STIM1 inactivation domain play an important role in pH_i_ sensitivity [223].

For what concerns hypoxia-mediated regulation of SOCs, intracellular acidification of primary aortic smooth muscle cells and HEK293 cells induced by hypoxia led to inhibition of SOCE by disrupting the electrostatic ORAI1/STIM1 binding and closing ORAI1 channel (Table 2 and Figure 2). Nonetheless, STIM1 remained associated with ORAI1 through the second binding site located between ORAI1’s intracellular N-terminal tail and STIM1’s STIM-ORAI activating region (SOAR), preventing the noxious hypoxia-mediated Ca^2+^ overload [92]. Therefore, intracellular hypoxia-mediated acidification might regulate SOCs activity by uncoupling ORAI11 and STIM1 and, consequently, reducing I_CRAC_ amplitude.

As ORAI/STIM mediate most of the Ca^2+^ signalling in cancer-induced acidosis, they play a key role in several pathological processes. In the cancer context, it was observed that SOCE is regulated by changes in extracellular pH, as acidification of tumour microenvironment suppressed both the carbachol-(CCH) and thapsigargin (TG)-mediated Ca^2+^ entry in neuroblastoma cells, while external alkalinisation increased both the CCH- and the TG-induced Ca^2+^-influx [93], therefore in accordance with the results obtained in non-cancer cells (Table 2 and Figure 2).

However, the work of Liu X. et al. of 2018 on triple-negative breast cancer cells (TNBCs) reported that hypoxia promoted the activation of Notch1 signalling, required for the upregulation of ORAI1 mRNA and protein levels in TNBCs (Table 2 and Figure 2). In addition, the upregulation of ORAI1 in hypoxia determined an increase in basal Ca^2+^ concentration and of thapsigargin-induced SOCE, which activated the downstream NFAT4 target, known to regulate the expression of cancer-related genes involved in its hallmarks [128]. The results of the same work showed that hypoxia enhanced invasion and TNBC migration via the Notch1/ORAI1/SOCE/NFAT4 pathway; therefore, ORAI1 and SOCE play a key role in promoting an aggressive phenotype. Same results were obtained in colon cancer cells by the same group of researchers, where potentiation of SOCE mediated by hypoxia-induced upregulation of ORAI1 determined the activation of NFATc3, enhancing hypoxia-induced invasion and angiogenesis in colon cancer cells [129]. The hypoxia-induced upregulation of SOCE components was also demonstrated in A549 and NCI-H292 non-small cell lung cancer cells, where nicotine treatment determined the upregulation of HIF-1α, which increased the expression of the SOCE components TRPC1, TRPC6 and ORAI1. This translated into potentiation of SOCE and calcium entry, promoting lung cancer cells’ proliferation [127].

Considering the acidosis and hypoxia-mediated effects presented, the inhibitory action of the tumour intra- and extracellular acidic pH on SOC channels seems to contradict the positive effect of SOCE on tumour progression. In particular, acidosis inhibition of SOCE does not correlate with the different studies demonstrating that ORAI/STIM and SOCE promote cancer development [225] and that also tumour acidosis and hypoxia support it by enhancing tumour cell migration, invasion and, therefore, its aggressive phenotype [40,226]. Possible explanations for the apparent counterproductive blockade of SOCE channels by the tumour’s acidic pH lie in the fact that Ca^2+^ signalling not only promotes tumour progression and development by potentiation of its hallmarks but also contributes to its suppression by enhancing processes such as cell death, senescence and autophagy [49]. In addition, ORAI members assembly to form different combinations of heteromeric Ca^2+^ release-activated channels (CRACs) and the ratio of each ORAI member determines specific I_CRAC_ current properties and CRAC effects [227,228]. Therefore, the acidic pH of the tumour microenvironment may differently regulate heteromeric CRACs. Another point to consider is the key role of SOCE in immune cell activation [229]. The requirement of Ca^2+^ entry for antitumour immunity might explain the inhibitory effect of acidic tumour microenvironment on SOCE, in order to decrease immune cells’ function and protect the tumour.

## 3. Concluding Remarks

The present review aimed to overview the crosstalk between major chemical components of the tumour microenvironment, hypoxia and acidic pH_e_, and Ca^2+^-permeable ion channels, summarizing the major Ca^2+^-mediated signalling pathways that are involved in hypoxia and acidic pH_e_ responses in cancer cells. We focused our attention mainly on not-voltage gated TRPs, SOCs as well as Piezo channels due to their role as polymodal sensors in the tumour microenvironment. In this perspective, the data presented showed that hypoxia has a positive effect on some TRP and ORAI1 channels (Figure 3), promoting their activation and their expression via different transcription factors. On the contrary, tumour acidosis modulation shows a higher variability, determining loss-of-function of specific Ca^2+^-permeable ion channels expressed not only by cancer cells but also by immune cells, or potentiation or activation of others.

Although the separate roles of tumour acidosis, hypoxia and Ca^2+^ signalling in cancer progression are well established, less is known about their crosstalk in this context, with the majority of works cited in this review performed on normal cell lines and with a focus on acidic pH_e_ effect on Ca^2+^-permeable ion channels’ current via electrophysiology experiments. In addition, investigation of intracellular acidification-mediated regulation of Ca^2+^ signals and the pH_i_-dependence of Ca^2+^-permeable channels in cancer is rare, although it is well known that intracellular pH plays a more decisive role in regulating various biological processes and that this value is highly influenced by the extracellular pH. Based on the established knowledge about Ca^2+^-dependent signalling pathways involved in tumour progression and on the information about the effects of hypoxia and acidosis on the activity and expression of different Ca^2+^-permeable channels mostly available from normal or tumourigenic cells studies, the present review critically presented and discussed this crosstalk to elucidate and hypothesize which of TRPs, ORAIs and PIEZO channels Ca^2+^-dependent signalling pathways could be activated or inhibited by hypoxia and tumour acidosis-mediated regulation of Ca^2+^-permeable ion channels in cancer cells and involved in tumour progression, for the identification of potential molecular identities to target. In particular, the putative synergistic relationship between hypoxia, tumour acidosis and low pH_e_-induced intracellular acidification and Ca^2+^ signals, the mechanisms of their interaction and their interdependence in tumours require further studies and clarifications, in order to fill the gap and promote a better understanding of the crosstalk between three major players in the cancer research field.

## Figures and Tables

**Figure 1 ijms-23-07377-f001:**
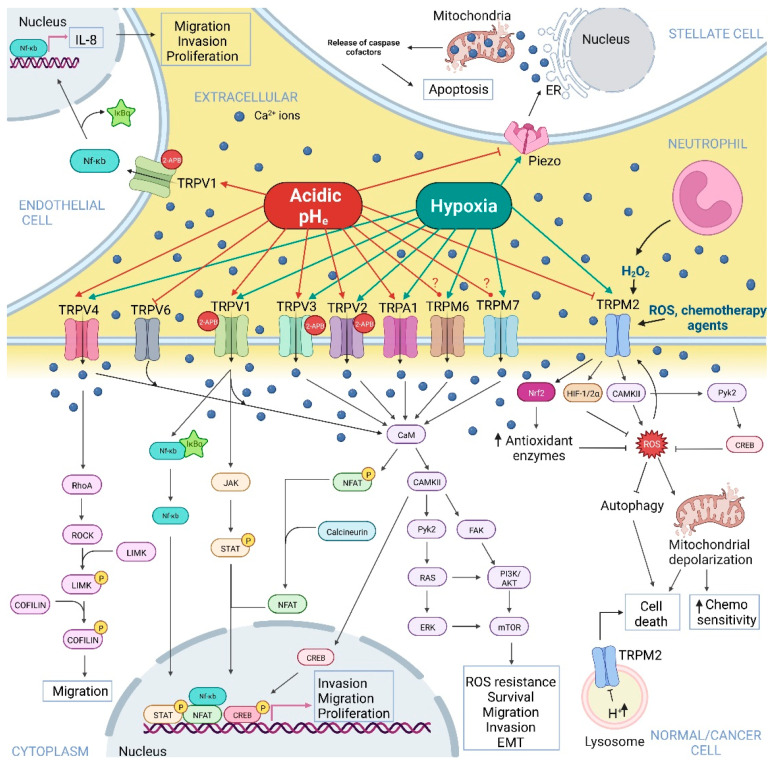
Overview of the effects of acidic pH_e_ and hypoxia on TRP and Piezo Ca^2+^-permeable channels. The positive or negative effects of hypoxia and acidic pH_e_ on the Ca^2+^-permeable channels were obtained from both normal and cancer cells-related studies, while the signalling pathways indicated were obtained uniquely from cancer cells-based investigations. The figure depicts Piezo-, TRPV-, TRPA1- and TRPM-mediated Ca^2+^-dependent signalling pathways activated or inhibited by acidic pH_e_ and hypoxia and linked to tumour progression. TRPV (TRPV1–4, 6), TRPA1, and TRPM (TRPM6, 7) expressed in cell cancer’s plasma membrane are differentially regulated by acidic pH_e_, being mostly activated by tumour acidosis, and transducing its signals to activate Ca^2+^-dependent downstream effectors, such as NF-κB, JAK/STAT, PI3K/AKT, NFAT, ERK, and LIMK. TRPA1 is also activated by hypoxia. These effectors promote tumour cell migration, invasion, proliferation, survival, mesenchymal phenotype, and chemoresistance. TRPV6 channels’ activity is inhibited by tumour acidosis, as TRPM2, which inhibition avoids induction of cancer cell death and reduces chemosensitivity. Piezo channels embedded in stellate cells’ plasma membrane are inhibited by acidic pH_e_, promoting stellate cells’ survival. TRPV1 activation in lymphatic endothelial cells promotes activation of NF-κB and upregulation of IL-8, a lymphangiogenic factor. CaM, calmodulin; CAMKII, Ca^2+^/calmodulin-dependent protein kinase II; Pyk2, protein tyrosine kinase 2; RAS, Rat sarcoma virus; ERK, extracellular signal-regulated kinase; FAK, Focal Adhesion Kinase; PI3K, phosphoinositide 3-kinase; AKT, protein kinase B; mTOR, mammalian target of rapamycin; NF-κB, nuclear factor-κB; JAK, Janus kinases; STAT, signal transducer and activator of transcription; NFAT, nuclear factor of activated T-cells; RhoA, Ras homolog family member A; ROCK, Rho-associated protein kinase; LIMK, LIM domain kinase; CREB, C-AMP response element-binding protein. The question mark indicates contradictory results in the literature. Created with BioRender.com, accessed on 20 June 2022.

**Figure 2 ijms-23-07377-f002:**
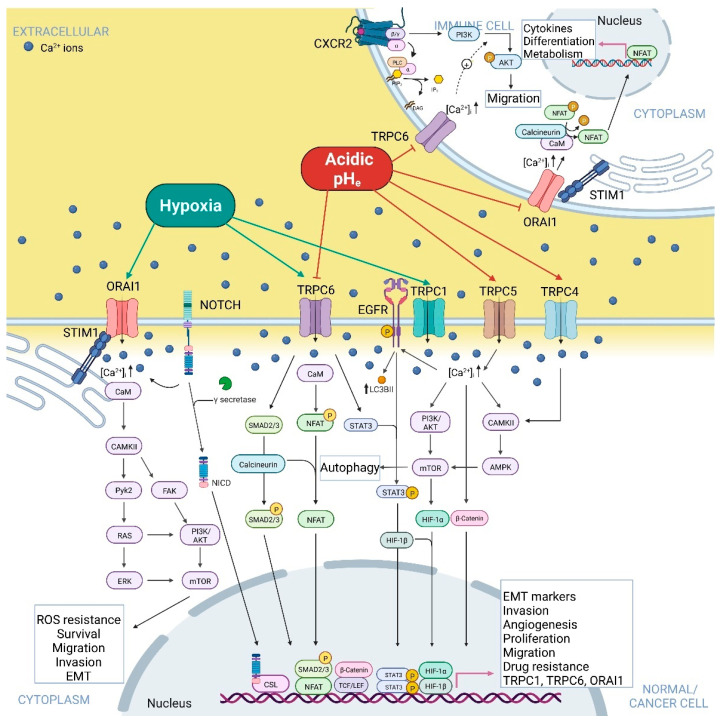
Overview of the effects of acidic pH_e_ and hypoxia on TRPC and ORAI Ca^2+^-permeable channels. The positive or negative effects of hypoxia and acidic pH_e_ on the Ca^2+^-permeable channels were obtained from both normal and cancer cell-related studies, while the signalling pathways indicated were obtained uniquely from cancer cell-based investigations. The figure depicts TRPCs- and SOCs-mediated Ca^2+^-dependent signalling pathways inhibited or activated by acidic pH_e_ or hypoxia and linked to tumour progression. TRPC (TRPC1, 4, 5) expressed in cancer cells’ plasma membrane is all activated by acidic pH_e_ or hypoxia, transducing their signals to activate Ca^2+^-dependent downstream effectors, such as SMAD2/3, NFAT, STAT3, HIF1, AMPK and β-catenin. These effectors promote tumour cell migration, angiogenesis, invasion, proliferation, mesenchymal phenotype and chemoresistance and the expression of TRPC1, via EGFR activation, and TRPC6 via Notch1 signalling pathway, in a mechanism of positive feedback regulation for both TRPC1 and TRPC6 channels. Immune cells expressing TRPC6 channels on plasma membrane show TRPC6’s activity that is inhibited by acidic pH_e_, reducing their migration. ORAI1 channels function in immune cells is also negatively affected by acidic pH_e_, impairing different processes needed for immune cells’ anti-tumour activity. Hypoxia promotes both ORAI1 expression, via Notch signalling pathway, and activation, leading to increased ROS resistance, migration, invasion, EMT and cell survival. CaM, calmodulin; CAMKII, Ca^2+^/calmodulin-dependent protein kinase II; Pyk2, protein tyrosine kinase 2; RAS, rat sarcoma virus; ERK, extracellular signal-regulated kinase; FAK, focal adhesion kinase; PI3K, phosphoinositide 3-kinase; AKT, protein kinase B; mTOR, mammalian target of rapamycin; NICD, Notch intracellular domain; CSL, CBF1, suppressor of hairless, Lag-1; NFAT, nuclear factor of activated T-cells; STAT, signal transducers and activators of transcription; EGFR, epidermal growth factor receptor; HIF-1, hypoxia-inducible factor 1. Created with BioRender.com, accessed on 20 June 2022.

**Figure 3 ijms-23-07377-f003:**
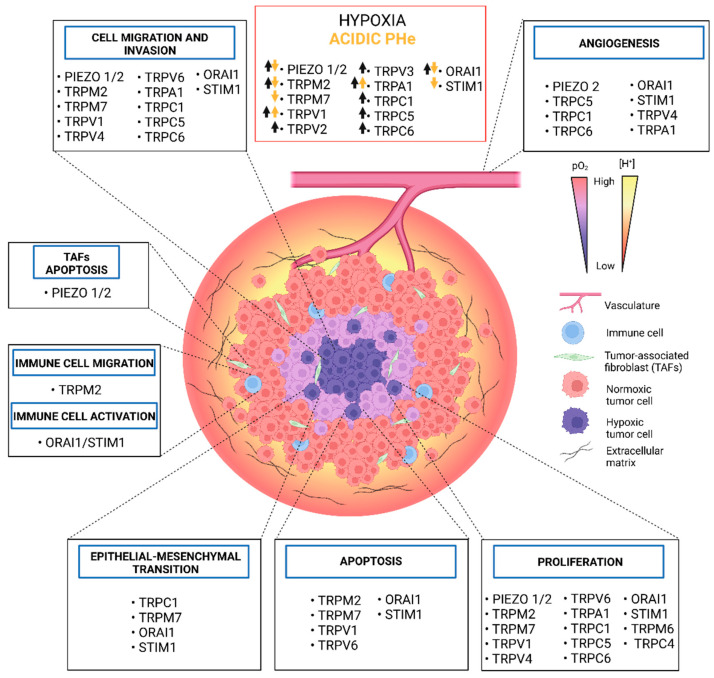
Schematic illustration of tumour microenvironment landscape. The increased tumour growth, the acidification of the extracellular space and aberrant vascularisation and limited O_2_ supply origin a tumour core that is hypoxic and acidic, with limited supply of oxygen and nutrients from the blood vessels. Peripheral tumour cells are located in regions with a higher extracellular pH, a result of proximity to blood vessels and the possibility to wash out acidic waste products. TRP, Piezo and SOCs channels expressed in cancer, immune and stromal cells are presented in the corresponding black boxes with indication of their involvement in different cancer hallmarks, such as proliferation, migration, invasion, angiogenesis, and epithelial–mesenchymal transition. The red box contains the up-to-date information regarding the effect (up arrows = positive effect; down arrows = negative effect) of hypoxia and acid pH_e_ on the activity and/or expression of calcium-permeable channels in cancer cells or tumour-associated cells.

## Data Availability

Not applicable.

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
