# Peer review of "Ca^2+^ Signalling and Hypoxia/Acidic Tumour Microenvironment Interplay in Tumour Progression"

_ijms, 2022, doi:10.3390/ijms23137377_

Round 1
Reviewer 1 Report
The Ms entitled “Ca2+ signaling and hypoxia/acidic tumour microenvironment: interplay in tumour progression” by Madelaine M. Audero, Natalia Prevarskaya and Alessandra Fiorio Pla review the current knowledge regarding the possible relationship between several calcium transporters, specifically TRP, SOC and PIEZO1 channels, and the acidic and hypoxic microenvironment typical of solid tumors. Many evidences are reported for a regulation of the above mentioned ion channels by oxygen levels and changes in the extracellular pH, as well as on their expression and relevance in the progression of tumor cell.
1) The Ms deals with a very important topic and it can be very useful for readers interested in the mechanism by which the tumor microenvironment regulates the development, spreading and progression of tumor cells. However, several points needs to be addressed by the authors to improve their Ms
1. 2) The authors reports many evidences for a regulation of the TRP, SOC and PIEZO1 channels by extracellular acidic pH and low oxygen levels. These evidences are taken from both normal and tumor cells, without a clear separation between the two. I suggest to stress much more the evidences coming from tumor cells, while down-regulating those coming from normal cells. This because the above mentioned regulations may involve signal transduction mechanisms that can be cell specific, and for this reason evidences coming from normal cells might not be applicable to tumor cells.
2. 3) The two physico-chemical factors taken into consideration, hypoxia and extracellular pH, are treated at a very different level of detail, with the information regarding the hypoxia-regulated ion channels much less present as compared to that regarding the extracellular pH. Just as an example, evidences for a regulation of PIEZO1 channels by hypoxia (see Wang et al., 2009 FASEB; Vandorpe et al., 2010 Plos One) is completely absent. It is also surprising that the authors do not cite and do not take into account data reported by previous work on the possible role of hypoxia modulated channels in the aggressiveness of brain tumors (Sforna et al., 2015 Frontiers in Cell Neurosci; Sforna et al., 2017; J Cell Physiol; Rosa et al., 2018). This literature should be cited.
3. 4) The are many grammatical errors and incomprehensible sentences throughout the Ms. Thus an English editing by an expert is needed.
4. 5) More evidence should be reported on the possible role of calcium signaling in the aggressiveness of tumor cells (see for example Catacuzzeno and Franciolini, 2018 Int J Mol SCi), as well as evidences reporting that calcium levels might be under the control of hypoxia and extracellular pH.
Author Response
We thank the reviewer for the constructive criticisms. To answer the major comments raised by the reviewer, we modified Fig.3 which is now just focused on cancer TME and only Hypoxia and acidic pHe effects on PIEZO, TRPs and SOC channels in cancer are reported. The reviewer also suggested to better develop the hypoxia-mediated role on Ca2+ cannels. According to this suggestion, we added specific paragraphs for the different channels in section 2. Moreover, we divided the previous Table1 into two new tables: New Table 1 focused in acidosis and New Table 2 summarizing the role of hypoxia. The English form has been extensively revised to make the manuscript easier to read.
The detailed answers to the reviewer's comments are as follow:
- “The authors reports many evidence for a regulation of the TRP, SOC and PIEZO1 channels by extracellular acidic pH and low oxygen levels. These evidence are taken from both normal and tumor cells, without a clear separation between the two. I suggest to stress much more the evidences coming from tumor cells,while down-regulating those coming from normal cells. This because the above mentioned regulations may involve signal transduction mechanisms that can be cell specific, and for this reason evidences coming from normal cells might not be applicable to tumor cells.”
We thank the reviewer for point out the discrepancy. Indeed, giving the fact that data on cancer cells is definitely less studied, much information regarding pH and hypoxia-mediated regulation were obtained from investigations on normal cells or other pathologies.
We agree with the reviewer that acidosis/hypoxia regulation may involve different signalling pathways between tumour and normal cells. To better clarify the literature knowledge on tumour cells, we modified Fig 3 which is now just focused on cancer TME and only Hypoxia and acidic pHe effects on PIEZO, TRPs and SOC channels in cancer are reported. Moreover, we specified in the conclusions that we took into account the information on pH- and hypoxia-mediated regulation present in publications carried out in normal or tumorigenic cells on one hand, and the established knowledge of calcium-dependent molecular pathways activated in cancer on the other. We critically presented and discussed this crosstalk to elucidate and hypothesize which of TRPs, ORAIs and PIEZO channels Ca2+-dependent signalling pathways could be activated or inhibited by hypoxia and tumour acidosis-mediated regulation of Ca2+-permeable ion channels in cancer cells and involved in tumour progression, for the identification of potential molecular identities to target.
- “The two physico-chemical factors taken into consideration, hypoxia and extracellular pH, are treated at a very different level of detail, with the information regarding the hypoxia-regulated ion channels much less present as compared to that regarding the extracellular pH. Just as an example, evidence for a regulation of PIEZO1 channels by hypoxia (see Wang et al., 2009 FASEB; Vandorpe et al., 2010 Plos One) is completely absent. It is also surprising that the authors do not cite and do not take into account data reported by previous work on the possible role of hypoxia modulated channels in the aggressiveness of brain tumors (Sforna et al., 2015 Frontiers in Cell Neurosci; Sforna et al., 2017; J Cell Physiol; Rosa et al., 2018). This literature should be cited.”
We agree with the reviewer about the differences in the description of hypoxia or extracellular pH effect on ion channels considered. To better clarify the role of hypoxia in Ca2+-permeable ion channels activity/expression, several paragraphs and references were added starting from section 2. Indeed we clarified for most of the channels hypoxia-mediated regulation. Moreover old Table1 is not split into two new tables: New Table 1 focused in acidosis and New Table 2 summarizing the role of hypoxia on PIEZO, TRP and SOC channels. Both tables include data from both normal and cancer cells
We thank the reviewer for pointing out the missing references concerning Piezo1 that have been now included in revised version (Wang et al., 2009 FASEB; Vandorpe et al., 2010 Plos One). We also included a recent review by Catacuzzeno and Sforna (2020).
- “The are many grammatical errors and incomprehensible sentences throughout the Ms. Thus, an English editing by an expert is needed.”
We extensively revised the manuscript and the English form to correct the typos and make sentences more clear.
- “More evidence should be reported on the possible role of calcium signaling in the aggressiveness of tumor cells (see for example Catacuzzeno and Franciolini, 2018 Int J Mol SCi), as well as evidence reporting that calcium levels might be under the control of hypoxia and extracellular pH.”
To better clarify the role of Ca2+ signalling in cancer and previous evidence in literature about Ca2+-signaling and acidic pH/hypoxia link, new paragraphs and references were added (lines 199-221; ref. 53).
Reviewer 2 Report
The present review will focus on Ca2+ permeable ion channels, with a major focus on TRP, SOC and PIEZO channels, that are modulated by tumor hypoxia and acidosis as well as the role of the altered Ca2+ signals on cancer progression hallmarks. The scientific collect is very interesting, however, some minor problems, as indicated below, should be addressed before the document can be considered for publication. This version of the manuscript is not enough complete.
Minor revision:
-English language and style are fine, moderate spell check is required to ensure that an international audience can clearly understand your text. In general, I suggest to review the style of the manuscript according to the guidelines of the journal.
-A class of ion channel implicated in cancer is transient receptor potential (TRP) channels. Since most TRP channels are permeable to Ca2+, increased TRP channel expression is expected to enhance intracellular Ca2+ levels and impact Ca2+ signaling. Indeed, overexpression of several TRP channel genes in various cancer types is compatible with a switch of cancer cells to an aberrant proliferative state. Related to the aberrant proliferation and the often hypoxic environment, most cancers are characterized by enhanced oxidative stress, which can be directly sensed by various TRP channels, including TRPM2.
Interestingly, TRPM2 is implicated in several physiological and pathological pathways involving oxidative stress, and its activation has been generally associated with a large increase in intracellular calcium. For example, TRPM2 activation causes a dramatic increase in intracellular calcium levels, which stimulates two distinct Ca2+-dependent K+ channels in melanoma: the large-conductance voltage-dependent BK channel, and the medium-conductance voltage-independent KCa3.1 channel. Both channels could modulate the cancer progression. I suggest to discuss this evidence, adding two recent references (DOI: 10.3390/ijms22168359; DOI: 10.1016/j.bbamcr.2020.118644) in the full text (Lines 314-334).
-I suggest to explain better the following sentence "This channel is upregulated in several cancers with a pro-tumour effect via Ca2+-dependent pathways. This effect may be explained by the protective role promoted by TRPM2 activation that act as a ROS sensor and promotes in turn activation of transcription factors involved the increase level of antioxidant (i.e. HIF-1/2a; CREB; NrF2)"
Pro-tutor effect?
-Recent pieces of evidence suggested that dysregulation of the endothelial Ca2+ machinery is crucial to support neovascularization and resistance to anti-angiogenic and chemotherapeutic drugs in a growing number of malignancies. I suggest to add this recent review, in which the mechanisms whereby aberrant expression and/or activity of endothelial TRP channels impact on tumor vascularization have been collected (DOI: 10.3389/fphys.2019.01618).
Author Response
Reviewer 2
We thank the reviewer for the constructive criticisms.
The detailed answers to the minor reviewer's comments are as follow:
- “English language and style are fine, moderate spell check is required to ensure that an international audience can clearly understand your text. In general, I suggest to review the style of the manuscript according to the guidelines of the journal.”
We extensively revised the manuscript to correct the typos and make sentences more clear respecting journal guidelines.
- “A class of ion channel implicated in cancer is transient receptor potential (TRP) channels. Since most TRP channels are permeable to Ca2+, increased TRP channel expression is expected to enhance intracellular Ca2+ levels and impact Ca2+ signaling. Indeed, overexpression of several TRP channel genes in various cancer types is compatible with a switch of cancer cells to an aberrant proliferative state. Related to the aberrant proliferation and the often hypoxic environment, most cancers are characterized by enhanced oxidative stress, which can be directly sensed by various TRP channels, including TRPM2. Interestingly, TRPM2 is implicated in several physiological and pathological pathways involving oxidative stress, and its activation has been generally associated with a large increase in intracellular calcium. For example, TRPM2 activation causes a dramatic increase in intracellular calcium levels, which stimulates two distinct Ca2+-dependent K+ channels in melanoma: the large-conductance voltage-dependent BK channel, and the medium-conductance voltage-independent KCa3.1 channel. Both channels could modulate the cancer progression. I suggest to discuss this evidence, adding two recent references (DOI: 10.3390/ijms22168359; DOI: 10.1016/j.bbamcr.2020.118644) in the full text (Lines 314-334). “
As suggested, we added and discussed References suggested by the reviewer (lines 448-453; reference 98 and 59).
- “I suggest to explain better the following sentence "This channel is upregulated in several cancers with a pro-tumour effect via Ca2+-dependent pathways. This effect may be explained by the protective role promoted by TRPM2 activation that act as a ROS sensor and promotes in turn activation of transcription factors involved the increase level of antioxidant (i.e. HIF-1/2a; CREB; NrF2)" Pro-tutor effect?
We agree with the reviewer about the poor clarity of the sentence that was modified accordingly (lines 439-447) and the role of TRPM2 in oxidative stress was better clarified.
- “Recent pieces of evidence suggested that dysregulation of the endothelial Ca2+ machinery is crucial to support neovascularization and resistance to anti-angiogenic and chemotherapeutic drugs in a growing number of malignancies. I suggest to add this recent review, in which the mechanisms whereby aberrant expression and/or activity of endothelial TRP channels impact on tumor vascularization have been collected (DOI: 10.3389/fphys.2019.01618).”
The interesting review has been now mentioned and the reference added (lines 199-202; ref. 51).

Round 2
Reviewer 1 Report
The Authors have performed the required changes